# SMCHD1 has separable roles in chromatin architecture and gene silencing that could be targeted in disease

Andres Tapia del Fierro [1,2], Bianca den Hamer[3], Natalia Benetti [1,2], Natasha Jansz [1,2], Kelan Chen[1,2], Tamara Beck[1], Hannah Vanyai[4], Alexandra D. Gurzau[1,2], Lucia Daxinger [5], Shifeng Xue [6,7], Thanh Thao Nguyen Ly[6,7], Iromi Wanigasuriya[1,2], Megan Iminitoff[1,2], Kelsey Breslin [1], Harald Oey[5], Yvonne D. Krom[3], Dinja van der Hoorn[3], Linde F. Bouwman [3], Timothy M. Johanson[1,2], Matthew E. Ritchie [1,2], Quentin A. Gouil [1,2], Bruno Reversade[7,8], Fabrice Prin[4], Timothy Mohun[4], Silvère M. van der Maarel [3], Edwina McGlinn [9,10], James M. Murphy [1,2,11], Andrew Keniry [1,2], Jessica C. de Greef [3] & Marnie E. Blewitt [1,2] ✉

The interplay between 3D chromatin architecture and gene silencing is incompletely understood. Here, we report a novel point mutation in the non-canonical SMC protein SMCHD1 that enhances its silencing capacity at endogenous developmental targets. Moreover, it also results in enhanced silencing at the facioscapulohumeral muscular dystrophy associated macrosatellite-array, *D4Z4*, resulting in enhanced repression of *DUX4* encoded by this repeat. Heightened SMCHD1 silencing perturbs developmental *Hox* gene activation, causing a homeotic transformation in mice. Paradoxically, the mutant SMCHD1 appears to enhance insulation against other epigenetic regulators, including PRC2 and CTCF, while depleting long range chromatin interactions akin to what is observed in the absence of SMCHD1. These data suggest that SMCHD1's role in long range chromatin interactions is not directly linked to gene silencing or insulating the chromatin, refining the model for how the different levels of SMCHD1-mediated chromatin regulation interact to bring about gene silencing in normal development and disease.

The importance of correct epigenetic regulation to normal development and differentiation has long been known. Modern genomic techniques have revealed sophisticated mechanisms behind epigenetic control, where DNA methylation, post-translational histone modifications, and chromatin conformation come together in a dynamic fashion to regulate gene expression. However, the complex interplay between different modes of epigenetic control, often between those considered to have opposing functions (like bivalent

[1]The Walter and Eliza Hall Institute of Medical Research, Melbourne, VIC, Australia. [2]The Department of Medical Biology, University of Melbourne, Melbourne, VIC, Australia. [3]Department of Human Genetics, Leiden University Medical Center, Leiden, Netherlands. [4]Crick Advanced Light Microscopy Facility, The Francis Crick Institute, London, UK. [5]Queensland Institute of Medical Research, Brisbane, QLD, Australia. [6]Department of Biological Sciences, National University of Singapore, Singapore, Singapore. [7]Institute of Molecular and Cell Biology, A*STAR, Singapore, Singapore. [8]Genome Institute of Singapore, A*STAR, Singapore, Singapore. [9]EMBL Australia, Monash University, Clayton, VIC, Australia. [10]Australian Regenerative Medicine Institute, Monash University, Clayton, VIC, Australia. [11]Drug Discovery Biology, Monash Institute of Pharmaceutical Sciences, Monash University, Parkville, VIC, Australia. ✉e-mail: blewitt@wehi.edu.au

chromatin[1]), has precluded straightforward interpretations. One approach to understanding such a complex biological problem is to unravel the factors involved in the process using unbiased genetic screens[2]. Indeed, genetic screening approaches have identified novel epigenetic regulators in yeast[3], plants[4], flies[5,6], and mammals.

We previously reported a sensitized in vivo screen in mice that paired *N*-ethyl-*N*-nitrosourea (ENU) mutagenesis with a variegating GFP transgene array to identify modifiers of transgene variegation, and therefore epigenetic regulation[7]. This screen led to the discovery of the *Smchd1* gene that encodes an epigenetic repressor, since shown to play a role in X chromosome inactivation[8,9], silencing of clustered gene families such as select imprinted clusters[10–12], the clustered protocadherins[10,11,13] and *Hox* genes[14,15]. By understanding more about SMCHD1 we can also learn more about how gene silencing works in each of these cases.

In addition to the developmental roles of SMCHD1 elegantly characterized in mice, SMCHD1 is also relevant to human disease. Heterozygous pathogenic variants have been found in the developmental disorders Bosma-arrhinia and microphthalmia syndrome[16,17] (BAMS) and facioscapulohumeral muscular dystrophy[18,19] (FSHD). BAMS is a craniofacial malformation syndrome and FSHD is a typically adult-onset debilitating progressive muscular dystrophy. FSHD is caused by the death of skeletal muscle cells due to aberrant expression of the germline and cleavage stage transcription factor DUX4. DUX4 is expressed when epigenetic silencing is relaxed by one of two distinct molecular mechanisms that both result in the same clinical outcome[20]. The *DUX4* gene is located within the *D4Z4* macrosatellite repeat on chromosome 4q35, which is normally comprised of 8–100 *D4Z4* tandem repeat units. FSHD type 1 is caused by repeat contraction to 1–10 units[21], wherein the number of repeat units is insufficient to trigger efficient silencing of the whole repeat array. FSHD type 2 is most commonly caused by loss-of-function mutations in SMCHD1, which results in de-repression of the *D4Z4* repeat array[18]. Patients who carry both SMCHD1 mutations and repeat contraction present with more severe disease, suggesting that SMCHD1 silencing activity is relevant to both subtypes of the disease[19], and that FSHD patients reside on a disease spectrum[22]. With SMCHD1 now known to contribute to at least two human diseases, it is of high interest to determine how SMCHD1 works and thus how its activity may be manipulated to treat disease.

SMCHD1 is a member of the structural maintenance of chromosomes (SMC) family. It interacts with chromatin through its hinge domain, which also facilitates homodimerisation[13,23,24]. Unlike the canonical SMC proteins, SMCHD1 is non-canonical in part because it possesses a GHKL ATPase domain[25–29]. The pathogenic variants identified in each of these two domains of the protein suggest that they are critical for SMCHD1 function. In BAMS, heterozygous missense variants restricted to the extended ATPase domain underlie disease[16,17,30]. By contrast, FSHD loss-of-function mutations occur across the length of the protein[31,32]. The differing mutation types and locations suggest that while loss of SMCHD1 function causes FSHD, in BAMS it may be due to altered protein function, in some cases mediated by gain of ATPase activity[16,30]. However, a comprehensive explanation as to why variants in SMCHD1 can cause such disparate phenotypes is currently unclear as one pathogenic variant has been identified in both BAMS and FSHD[33].

Recent work has begun to reveal both how SMCHD1 functions at, and is recruited to, the chromatin. For the inactive X chromosome, SMCHD1 recruitment is dependent on the polycomb repressive complex 1 (PRC1) catalyzed mark, histone 2A lysine 119 ubiquitination[34,35] (H2AK119ub). Somewhat like the canonical SMC protein complex, Cohesin, upon recruitment to target loci, SMCHD1 is involved in mediating long-range chromatin interactions[14,36,37]. SMCHD1 also appears to insulate against the occupancy and effect of other epigenetic regulators, such as CTCF and polycomb repressive complex 2 (PRC2)[13,14,36,37]. To date, our understanding of SMCHD1 function has largely been informed by the study of systems upon complete loss of SMCHD1 protein. While these studies have been informative, they are limited in that they reflect a complete loss-of-function scenario of the SMCHD1 protein and therefore cannot appropriately model all aspects of disease that arise due to *SMCHD1* perturbation.

In this study, we report a new *Smchd1* missense mutant that was identified in the same ENU mutagenesis screen that led to the original discovery of *Smchd1*. In this screen, mutations were named Modifiers of Murine Metastable Epialleles (Mommes)[7,38], with the original report detailing the *Smchd1* loss-of-function *MommeD1* mutation[7]. Here we describe the effects of the *MommeD43 Smchd1* mutation on gene expression, development, and chromatin architecture. These studies reveal that the *MommeD43* variant produces a gain-of-function effect on the expression of critical SMCHD1 targets, including in a mouse model of FSHD where the *MommeD43* variant can partially rescue *DUX4* silencing. Interestingly, our data on chromatin architecture and insulation suggest that SMCHD1's role in regulating long-range chromatin interactions is not required for silencing and is divorced from its role in chromatin insulation. Therefore, this study suggests an attractive starting point for SMCHD1 modulation in FSHD treatment and expands our understanding of how the multiple layers of SMCHD1-dependent chromatin regulation interact to elicit epigenetic silencing.

## Results

### MommeD43 is a Smchd1 mutant with increased transgene array silencing activity

In our ENU mutagenesis screen looking for epigenetic modifiers of transgene silencing, we identified a mutant line of mice, called *MommeD43*. The system used a transgene array of 11 units containing a GFP gene in a tandem repeat, directed to express in erythrocytes[39] (Fig. 1a). The *MommeD43* mutation caused a significant decrease in the proportion of erythrocytes expressing the GFP transgene as measured by flow cytometry, indicating enhanced silencing (Fig. 1a). The mutation was mapped using positional cloning in conjunction with linkage analysis utilizing SNP chip technology to mouse chromosome 17. Further refinement by whole exome sequencing revealed a cytosine to adenosine mutation in exon 15 of the *Smchd1* gene, translating as an Alanine 667 to Glutamic acid conversion (A667E) in the extended ATPase domain of SMCHD1[26] (Fig. 1b). We performed whole-genome Nanopore long-read sequencing of mutant and wild-type samples for additional investigation of both coding and non-coding variants. We confirmed that the exon 15 variant in *Smchd1* was the only exonic variant within the linked interval. There were 112 other single nucleotide variants in the interval, 9 in potential regulatory regions; however, no variants are likely detrimental (Supplementary Data 1). Interestingly, *MommeD43* has the opposite effect on transgene silencing to the nonsense mutation in *Smchd1* identified in the MommeD1 strain[7,8]. Together these data suggest that *MommeD43* could be a gain-of-function mutation in *Smchd1*.

Homozygous *Smchd1* null animals show complete female embryonic lethality due to failure of X chromosome inactivation and reduced viability in males, dependent on strain background[8,11]. By contrast, *Smchd1^MommeD43/+^* and *Smchd1^MommeD43/MommeD43^* mice of both sexes are viable and fertile, with no observed transmission ratio distortion of each genotype from heterozygous intercrosses (Supplementary Fig. 1).

To explore how the *MommeD43* mutation in *Smchd1* may influence SMCHD1 function, we first considered whether the mutation results in increased levels of SMCHD1 protein. We found no differences in SMCHD1 protein levels when measured by western blot or flow cytometry (Fig. 1c, Supplementary Fig. 2). We next wanted to test whether the *MommeD43* mutation resulted in altered chromatin occupancy by chromatin immunoprecipitation followed by next-generation sequencing (ChIP-seq). Lacking antibodies against SMCHD1 that were of high enough quality for ChIP-seq, we developed

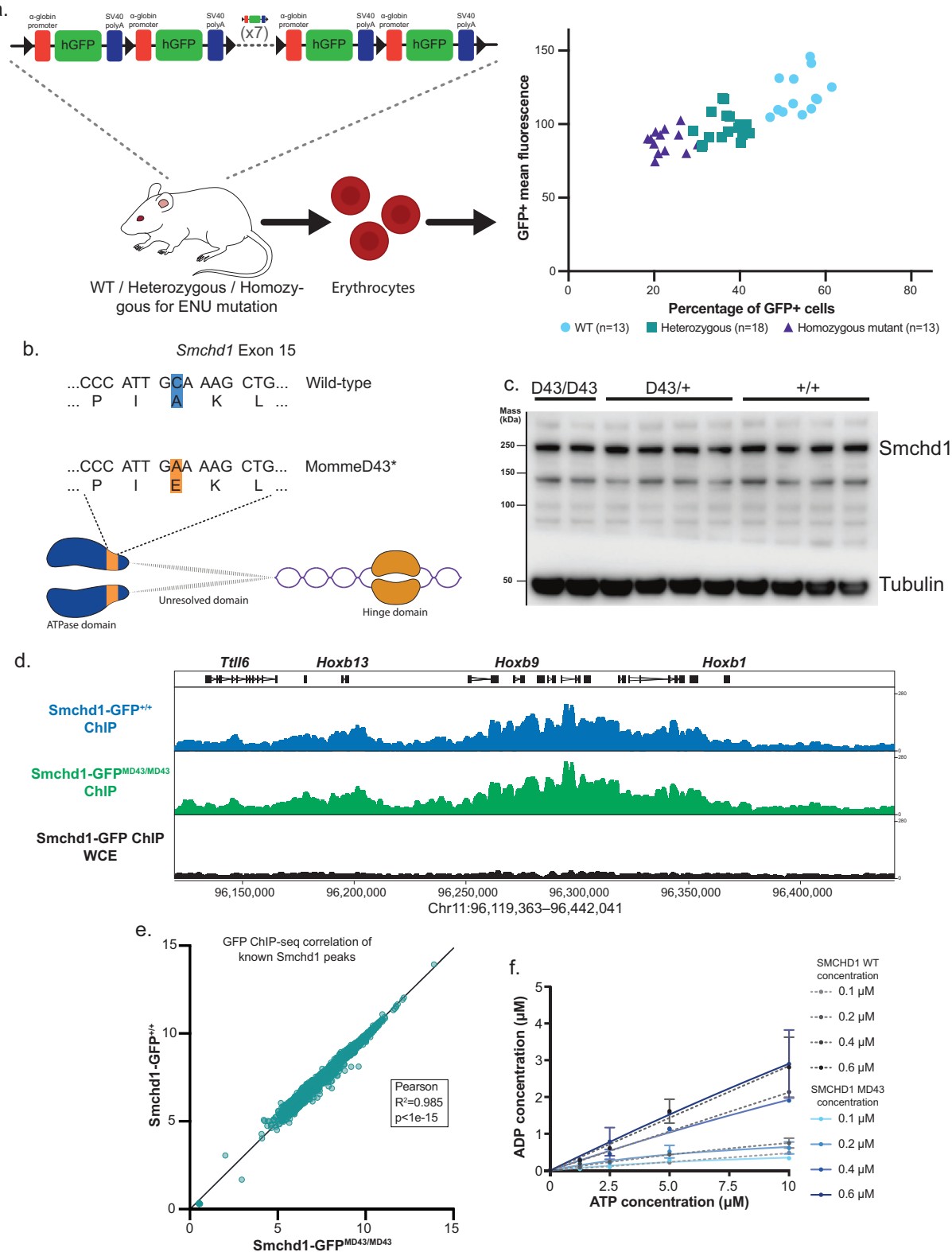

a mouse line carrying the *MommeD43* mutation by CRISPR/Cas9 editing of our previously described *Smchd1-GFP* fusion mouse model[14], allowing immunoprecipitation of SMCHD1 with an antibody to GFP. Both the *Smchd1^MommeD43-GFP* and *Smchd1^MommeD43* alleles are used in the remainder of this study. We have performed most genomic experiments in the C57BL/6J CRISPR allele to be consistent with these ChIP-seq data, while we have performed most embryo studies with the FVB/NJ ENU allele because the known vigor of the FVB/NJ background

makes such studies far more feasible. Importantly, whenever we have been able to use both the ENU and the CRISPR mutant allele, we have obtained the same results, as detailed in later sections and summarized in Supplementary Table 1.

We tested whether the *MommeD43* mutation altered SMCHD1 chromatin occupancy by performing anti-GFP ChIP-seq in *Smchd1^GFP/GFP* and *Smchd1^MommeD43-GFP/MommeD43-GFP* primary female neural stem cells (NSCs), alongside a whole cell extract (WCE) input control. We elected

**Fig. 1 | MommeD43 is a *Smchd1* mutant with increased transgene array silencing activity. a** Diagram of ENU screen experiment (left) and results (right). Erythrocytes from mice with an 11-unit GFP transgene repeat array carrying heterozygous or homozygous *MommeD43* mutations and wild-type littermate controls were analyzed by FACS to measure GFP transgene expression levels. **b** Schematic representation of murine SMCHD1 (resolved ATPase and hinge domains linked by flexible still unresolved domain) and the location of the MommeD43 mutation in its structure (orange). **c** Western blot of SMCHD1 in *Smchd1^MommeD43/MommeD43^*, *Smchd1^MommeD43/+^*, and *Smchd1^+/+^* cells showing no noticeable change in SMCHD1 levels (representative image of two independent experiments). **d** ChIP-seq genome browser tracks of the *Hoxb* cluster locus showing GFP ChIP in primary NSCs with

endogenous SMCHD1-GFP fusion protein, compared to the whole cell extract (WCE) input control. This region is heavily marked by SMCHD1 and *MommeD43* does not alter localization. On top are indicated a few genes in the locus for reference. **e** Scatter plot of log2-transformed normalized GFP ChIP-seq counts in *Smchd1^GFP/GFP^* and *Smchd1^MommeD4-GFP3/MommeD43-GFP^* NSCs around (±5 kb) previously published peaks[14]. The Pearson coefficient indicates very high positive correlation showing no noticeable changes in SMCHD1 DNA binding sites, two-tailed *p* value. **f** ATPase assay using recombinant purified wild-type murine SMCHD1 extended ATPase domain (grayscale) compared with the *MommeD43* mutant equivalent (blue). Mean ± SD, *N* = 3 per concentration per protein. *MommeD43* abbreviated to *MD43*.

to study NSCs as these are a cell type in which we had extensive data on *Smchd1* which is useful for comparison[14]. We found a very high correlation of SMCHD1 levels within ±5 kb of known SMCHD1 binding sites (2840 peaks[14]) between the wild-type and *MommeD43* mutant cells (Fig. 1e, Pearson *R* > 0.99), such as the *Hoxb* cluster (Fig. 1d). These results suggest that *MommeD43* does not alter chromatin localization of SMCHD1. In summary, these data indicate that the *MommeD43* mutation does not alter the levels of SMCHD1 protein, nor its binding to target loci, suggesting that the mutation may instead alter protein function.

## The *MommeD43* mutation does not alter the conformation or activity of the extended ATPase domain

The *MommeD43* mutation is located in the extended ATPase domain of SMCHD1, which contains a catalytically active GHKL ATPase[25,26]. To test if the *MommeD43* mutation alters the conformation of this region, we analyzed the structure and activity of the recombinant extended ATPase domain expressed and purified from insect cells, as we have previously done[26]. The *MommeD43* mutant protein had no detectable change in ATPase activity (Fig. 1f). Using small-angle X-ray scattering (SAXS), we determined that the *MommeD43* mutant extended ATPase domain has the same gross topology as the wild-type protein (Supplementary Fig. 3, Supplementary Table 2). Therefore, we cannot attribute altered SMCHD1 function to a change in enzymatic activity or conformation. However, both of these results were obtained using the recombinant extended ATPase domain and therefore do not exclude the possibility that the mutation may confer changes to ATPase activity or conformation in the context of the full-length protein.

## *MommeD43* has a gain-of-function effect on *Hox* gene silencing and skeletal development

We next investigated the effect of the *MommeD43* mutation in development. SMCHD1 has a known role in *Hox* gene regulation and subsequent skeletal development[14]. We have previously observed SMCHD1 enriched at all four *Hox* clusters, which we observe again in both the control and *MommeD43* mutant samples in the ChIP-seq data presented here (Fig. 1d and Supplementary Fig. 4). Loss of *Smchd1* causes a posterior homeotic transformation at thoracic vertebra 13 (T13), consistent with the observed failure in posterior *Hox* gene silencing. To determine the effects of MommeD43 on skeletal morphology, we examined whole-mount skeletal preparations from embryonic day (E) 17.5 embryos. We found that the *MommeD43* mutation (ENU mutant allele) in *Smchd1* resulted in an anterior homeotic transformation with additional effects on rib formation. We divided these effects into three distinct and independent phenotypes (Fig. 2a, detailed scoring in Supplementary Fig. 5). First, a fusion of the ribs of the first two thoracic elements was observed with complete penetrance in homozygous mutants (*Smchd1^MommeD43/MommeD43^*, 12/12 embryos analyzed) and partial penetrance in heterozygous mutants (*Smchd1^MommeD43/+^*, 3/17; Fig. 2b). The second, partially-penetrant phenotype was an extra sternal rib attachment, where the rib from the 8th thoracic element connected to the sternum instead of being the first false rib, as in wild-type embryos (9/12 in *Smchd1^MommeD43/MommeD43^*, 12/17

in *Smchd1^MommeD43/+^*, Fig. 2c). The last phenotype found in some embryos was the presence of a well-formed rib from the vertebral element posterior to T13 (5/12 in *Smchd1^MommeD43/MommeD43^*, 1/17 in *Smchd1^MommeD43/+^*, Fig. 2d). In all cases this expansion of thoracic count was accompanied by a reduction in lumbar element number from 6 to 5, with no change in overall total vertebral number of the animal, indicating this phenotype represents an anterior homeotic transformation. Collectively, these phenotypes point to dysregulation of *Hox* gene expression and are consistent with *MommeD43* being a gain-of-function mutation in *Smchd1*.

To investigate *Hox* dysregulation directly, we dissected tailbud tissue that harbors progenitors of the vertebral column from *Smchd1^+/+^* and *Smchd1^MommeD43/MommeD43^* embryos at 8 somites (E8.5) and performed RNA-sequencing. In contrast to the upregulation of posterior *Hox* genes observed in *Smchd1* null tail buds at E9.5 found previously[14], here we see a mild decrease in multiple posterior *Hox* genes (Fig. 2e, Supplementary Data 2). We saw the largest effect on *Hoxa6* expression at the 8-somite stage (log2FC −1.3; *n* = 3 pairs of biological replicates), where the *Smchd1^MommeD43/MommeD43^* embryos had a decrease in expression, suggesting that *MommeD43* is causing a slight delay in the activation of *Hoxa6*. Interestingly, this delay may be sufficient to explain the T1-to-T2 rib fusion phenotype, as an identical phenotype was observed in *Hoxa6* null embryos[40], while T3-to-T4 rib fusion is observed upon *Hox6* paralogous deletion[41]. In a similar manner, the L1-to-T homeotic transformation might be explained by an activation delay of the second most affected gene, *Hoxd10*. *Hox10* paralogous deletion causes a lumbar to thoracic homeotic transformation in all lumbar vertebrae, but deletion of all paralogues except for one allele of *Hoxa10* causes a bilateral L1-to-T homeotic transformation[42]. Both *Hoxa6* and *Hoxd10* have clear SMCHD1 enrichment, as was observed for the entirety of all four *Hox* clusters (*Hoxb* in Fig. 1d, *Hoxa* and *Hoxd* in Supplementary Fig. 4). Given the upregulation of posterior *Hox* genes observed in *Smchd1* null tail buds at E9.5 found previously[14], these data are consistent with the *MommeD43* mutation causing a gain-of-function effect on SMCHD1.

## The *MommeD43* mutation does not recapitulate morphological changes observed in BAMS

BAMS is a rare human craniofacial malformation syndrome that causes partial or total loss of the nose and sense of smell, along with reduced eye size and other head morphological abnormalities (MIM:603457). It is caused by heterozygous missense variants in SMCHD1's extended ATPase domain, many of which are proven or predicted gain-of-function variants[16,30], although this remains a matter of debate[17,43]. Even though the equivalent A667E variant has never been observed in BAMS patients, the effects of *MommeD43* on SMCHD1 on transgene silencing, *Hox* gene expression, and skeletal patterning are consistent with a gain-of-function. This led us to examine craniofacial morphology and development in the *MommeD43* mutants.

We collected E14.5 embryos and analyzed them by High-Resolution Episcopic Microscopy (HREM). After reconstructing a high-resolution three-dimensional model of the embryos' morphology, we quantitated key craniofacial measurements (Fig. 3a–h) and

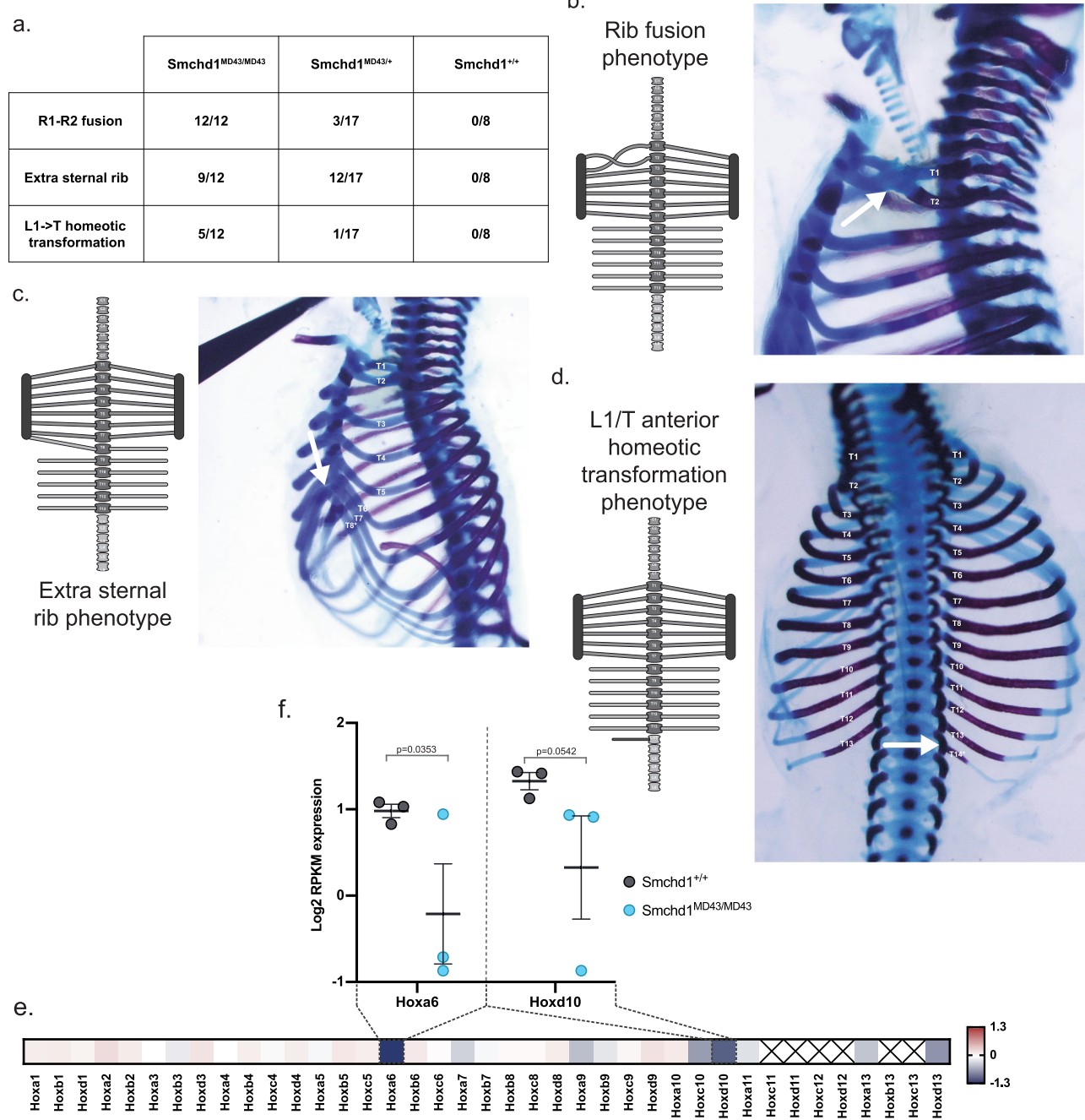

**Fig. 2 | *MommeD43* has a gain of function effect on *Hox* gene silencing and skeletal development. a** Scoring of the three observed skeletal phenotypes in *Smchd1^MommeD43/MommeD43^* and *Smchd1^MommeD43/+^* embryos (ENU mutant allele), *n* = 12, 17, and 9 for homozygous, heterozygous, and wild-type embryos. **b–d** Diagrams of the skeletal defects (left) of the three distinct phenotypes observed in *Smchd1^MommeD43/MommeD43^* E17.5 embryos with corresponding representative images (right). The white arrows point to the defect. **e** Heatmap of the log2 fold change between somite-paired *Smchd1^MommeD43/MommeD43^* and *Smchd1^+/+^* (ENU mutant allele) E8.5 embryos to account precisely for developmental stage. The values used are the mean expression of three biological replicates measured by RNA sequencing. **f** Plot of the normalized log2 expression values for the two *Hox* genes most affected by *MommeD43*. Each circle represents an individual biological replicate. Bars are mean ± SEM, *n* = 3, displayed *p* value of *F* test to show significantly increased variance in *MD43*. *MommeD43* abbreviated to *MD43*.

---

found minimal differences. A slight widening of the nasal capsule in *Smchd1^MommeD43/+^* embryos was the only parameter found to be significantly altered in mutants. Thus, we decided to test if *MommeD43* might affect gene expression during the development of the nose and snout. For this purpose, we dissected the frontonasal prominence (FNP, Fig. 3i) of somite-matched pairs of *Smchd1^MommeD43/MommeD43^* and *Smchd1^+/+^* E10.5 embryos (ENU mutant allele) and performed RNA-sequencing. Using duplicate somite-matched pairs at 29 somites, we found 56 differentially expressed genes (False Discovery Rate

(FDR) < 0.05, sex chromosomes excluded, Fig. 3j). Of these, 53 were downregulated in *Smchd1^MommeD43/MommeD43^* embryos which is consistent with the *MommeD43* mutation conferring better silencing capacity to SMCHD1 (Supplementary Data 3). Gene Ontology analysis showed that 7 of the 11 most significantly affected biological processes are directly related to development (Fig. 3k, corrected *p* value < 6 × 10^−9^). These data suggest that while the *MommeD43* mutation does not cause gross craniofacial abnormalities, it likely influences developmental gene expression programs that could be captured in the RNA-sequencing

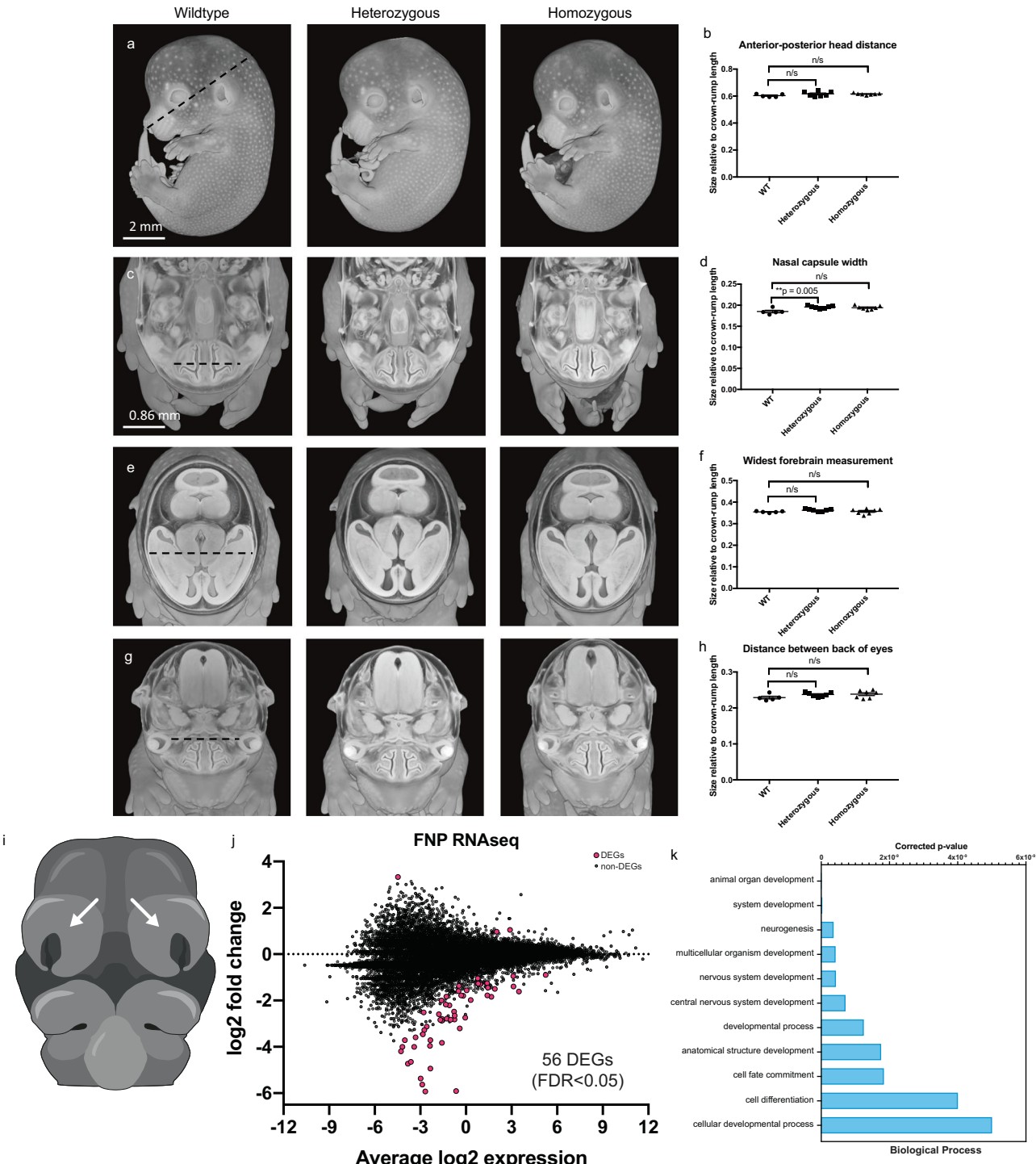

**Fig. 3 | The *MommeD43* mutation alters gene expression in the frontonasal prominences but does not recapitulate morphological changes observed in BAMS. a**–**h** Cutaways of three-dimensional renderings of E14.5 embryos imaged by HREM oriented to measure various craniofacial features (**a**, **c**, **e**, **g**) and graphs detailing the measurements normalized to embryo crown-rump length (**b**, **d**, **f**, **h**). n/s = not significant, **\**p = significant adjusted two-tailed *p* value (unpaired *t* test, Benjamini–Hochberg correction for multiple testing). *N* = 5–7 per genotype. Scale bar in **a** = 2 mm; scale bar in **c** = 0.86 mm and relates to **e**, **g**. **i** Diagram of E10.5 embryo. Arrow points to the FNP collected for RNA sequencing. **j** MD plot of log2

fold change of normalized RPKM counts of gene expression in FNP tissue from two somite-matched pairs of *Smchd1^MommeD43/MommeD43^* and *Smchd1^+/+^* (ENU mutant allele) E10.5 embryos showing 53 downregulated and 3 upregulated genes by *MommeD43* (56 total DEGs, FDR < 0.05). **k** Gene Ontology pathway analysis of the 11 main biological processes affected by the 37 uniquely mapped genes recognized by the GO platform out of the 56 DEGs shown in **j**. Bars show *p* values corrected for multiple testing (Fisher's exact *t* test, Benjamini–Hochberg correction). *MommeD43* abbreviated to *MD43*.

analysis. Interestingly, when the same region is harvested from *Smchd1^MommeD1/MommeD1* embryos, we observe upregulation of a restricted set of genes representing known SMCHD1 targets e.g. clustered protocadherins and genes from the *Snrpn* imprinted cluster (Supplementary Data 4), rather than the set of developmental genes downregulated in the *MommeD43* samples. Of the 56 differentially expressed genes in the *MommeD43* FNPs, 20 were nearby SMCHD1 binding sites in NSCs (peaks ±5 kb), a significant over-representation (Chi-square test of independence $p < 1 \times 10^{-26}$). Although FNPs and NSCs are two different tissues, these data suggest that the differentially expressed genes can be direct SMCHD1 targets, and are more sensitive to the *MommeD43* mutation than total loss of SMCHD1.

We next used an established assay[16,30] in which we microinjected *SMCHD1* cDNA into *Xenopus* to assess the effect of BAMS mutations in SMCHD1 on craniofacial development (Supplementary Fig. 6a). In this system, BAMS mutations result in a smaller eye phenotype, independent of whether we could detect an increase in ATPase activity in recombinant protein[16,30]. Here we compared the effect of the *MommeD43* mutation introduced into the human cDNA with that of a BAMS variant, W342S, with wild-type SMCHD1 and uninjected tadpoles as controls. While W342S mutant SMCHD1 resulted in a smaller eye, the *MommeD43* SMCHD1 behaved exactly as wild-type (Supplementary Fig. 6b, c). Taken together with the mouse embryology and the normal nasal morphology, these data suggest that the *MommeD43* variant does not accurately model BAMS in mice or frogs.

## *MommeD43*'s improved repeat-silencing capability offers therapeutic potential for FSHD

Since SMCHD1 is required for proper *DUX4* silencing, and the *D4Z4* repeat has a structure somewhat reminiscent of the GFP transgene array used in the ENU mutagenesis screen, we next examined whether *MommeD43* would also provide more efficient silencing of *DUX4*. To test this hypothesis, we crossbred heterozygous *Smchd1^MommeD43/+* mice (ENU mutant allele) with hemizygous *D4Z4-2.5* mice carrying a human transgene consisting of 2.5 *D4Z4* repeat units cloned from genomic DNA of an FSHD1 individual[44] (Fig. 4a). This approach was necessary because *D4Z4* is a primate-specific repeat[45]. We previously showed that murine SMCHD1 represses the human *D4Z4* transgene in these mice[46]. Thus, the *D4Z4-2.5* mouse model is suitable to evaluate the effect of *MommeD43* on *DUX4* expression and on the chromatin structure of the transgene.

The *D4Z4-2.5/Smchd1^MommeD43/+* mice exhibited Mendelian genotype and sex distributions (Supplementary Fig. 7a, b). They appeared healthy up to at least two months of age when they were sacrificed for this study. Males showed a slight increase in weight and females were comparable to the other measured genotypes (Supplementary Fig. 7c).

To determine whether *MommeD43* provides more efficient silencing of *DUX4* in vivo than wild-type SMCHD1, we first quantified the transcript levels of *DUX4*, *Wfdc3* (a murine target of human DUX4[44]), and *Smchd1* in three different skeletal muscles of *D4Z4-2.5/Smchd1^+/+* and *D4Z4-2.5/Smchd1^MommeD43/+* mice (gastrocnemius, quadriceps, triceps). *DUX4* expression levels were low and highly variable (Cq values varied between 34 and 38, while Cq values for *Gapdh* and *Rpl13a* were around 14 and 18, respectively), as previously reported for *D4Z4-2.5* mice[46], prohibiting the detection of consistent differences between *D4Z4-2.5/Smchd1^+/+* and *D4Z4-2.5/Smchd1^MommeD43/+* mice (Supplementary Fig. 8a). In line with this, we observed no changes in *Wfdc3* transcript levels (Supplementary Fig. 8b). As expected, we saw no change on *Smchd1* transcript levels (Supplementary Fig. 8c) or SMCHD1 protein levels (Supplementary Fig. 8d).

*DUX4* expression is typically higher in muscle cell cultures and non-muscle tissues of *D4Z4-2.5* mice than muscle tissue from the same animals as DUX4 blocks myogenesis in murine myocytes[46]. So, we turned to muscle cell cultures established from the extensor digitorum longus (EDL) muscle and several non-muscle tissues (cerebellum,

heart, spleen, thymus, fibroblast cell cultures). *DUX4* transcript levels were significantly decreased in both proliferating myoblast cultures and differentiating myotube cultures (Cq values between 29 and 33), in cerebellum (Cq values of 31-33) and in spleen (Cq values between 31 and 33) of *D4Z4-2.5/Smchd1^MommeD43/+* mice relative to samples from *D4Z4-2.5/Smchd1^+/+* animals (Fig. 4b, myotube differentiation confirmed by *Mef2c* levels, Supplementary Fig. 8k). In line with this, *Wfdc3* levels were also lower in EDL cultures established from *D4Z4-2.5/Smchd1^MommeD43/+* mice as well as in spleen (Fig. 4c). SMCHD1 transcript and protein levels were not affected in any of the studied cells or tissues (Fig. 4d, Supplementary Fig. 8c, d, g, h). Although murine *Wfdc3* was previously reported to be a human DUX4 target, it was not verified in all tissues[44]. *Wfdc3* levels were unchanged in cerebellum, which might suggest tissue-dependent responses to *DUX4*. We observed no changes in expression levels of either *DUX4* or *Wfdc3* in heart, thymus, or fibroblast cultures (Supplementary Fig. 8e, f).

As the exact mechanism of SMCHD1-enforced silencing of *DUX4* in skeletal muscle is unknown, we focused on known chromatin changes of the *D4Z4* repeat in FSHD. In humans, silencing of the *D4Z4* repeat is achieved by DNA methylation and repressive chromatin marks such as histone 3 lysine 9 trimethylation (H3K9me3) and histone 3 lysine 27 trimethylation (H3K27me3)[47–49]. Since *D4Z4* hypomethylation is observed in all tissues tested in carriers of a pathogenic variant in SMCHD1, we tested *D4Z4* methylation in tail DNA, as we previously did for *D4Z4-2.5* mice with a *Smchd1* nonsense mutation[46] and found *D4Z4* repeat hypomethylation. In tail DNA we observed no difference in DNA methylation levels at the DR1 or FasPas regions (Supplementary Fig. 8i), which are well-characterized regions within and just distal to the *D4Z4* repeat[50,51] respectively. Next, we measured the permissive chromatin mark H3K4me2 and the repressive histone marks H3K9me3 and H3K27me3 at the *D4Z4* repeat in both fibroblast cultures and spleen tissue. In individuals with FSHD, reduced H3K9me3 levels are found[49], while H3K27me3 levels are specifically increased in FSHD2 patients[52]. Although we did not observe any significant differences, H3K4me2 levels were slightly decreased in *D4Z4-2.5/Smchd1^MommeD43/+* mice compared with *Smchd1* wild-type counterparts, while H3K9me3 and H3K27me3 levels were higher. As a result, the chromatin compaction score (H3K9me3 level corrected for H3K4me2) was significantly increased in the *MommeD43* samples (Fig. 4e). This score is reduced in fibroblasts and myoblasts of individuals with FSHD[53] and in fibroblasts of *D4Z4-2.5* mice with the *Smchd1^MommeD1* (null) mutation[46]. This score was introduced to circumvent the correlation calculation problem of ChIP-qPCR experiments studying repetitive regions.

Our results suggest that *MommeD43* affects *DUX4* levels in specific tissues, perhaps by modulating the chromatin structure of the *D4Z4* repeat transgene. *MommeD43*'s more efficient silencing of DUX4 is not due to altered SMCHD1 binding to the transgene (Supplementary Fig. 8j), consistent with our genome-wide data. As mouse and human SMCHD1 are highly homologous, *MommeD43* may act as a hypermorphic variant in humans and may offer therapeutic potential for individuals with FSHD.

## The *MommeD43* mutation divorces the role of SMCHD1 in chromatin conformation from its role in gene silencing and chromatin insulation

Our data presented so far are consistent with *MommeD43* being a gain-of-function mutation. Since we have previously shown the effects of loss of SMCHD1 on gene expression, chromatin conformation, and histone marks in NSCs[14] we examined these features in *Smchd1^MommeD43-GFP/MommeD43-GFP* NSCs to define the mechanism behind the apparent gain-of-function in direct comparison to our existing *Smchd1* null data. NSCs provide a readily expandable source of uniform cells for genomic experiments, which would be far more challenging using dissected embryonic tissues due to the limited amount of tissue and low frequency of somite-matched embryos (e.g. FNPs and PSM).

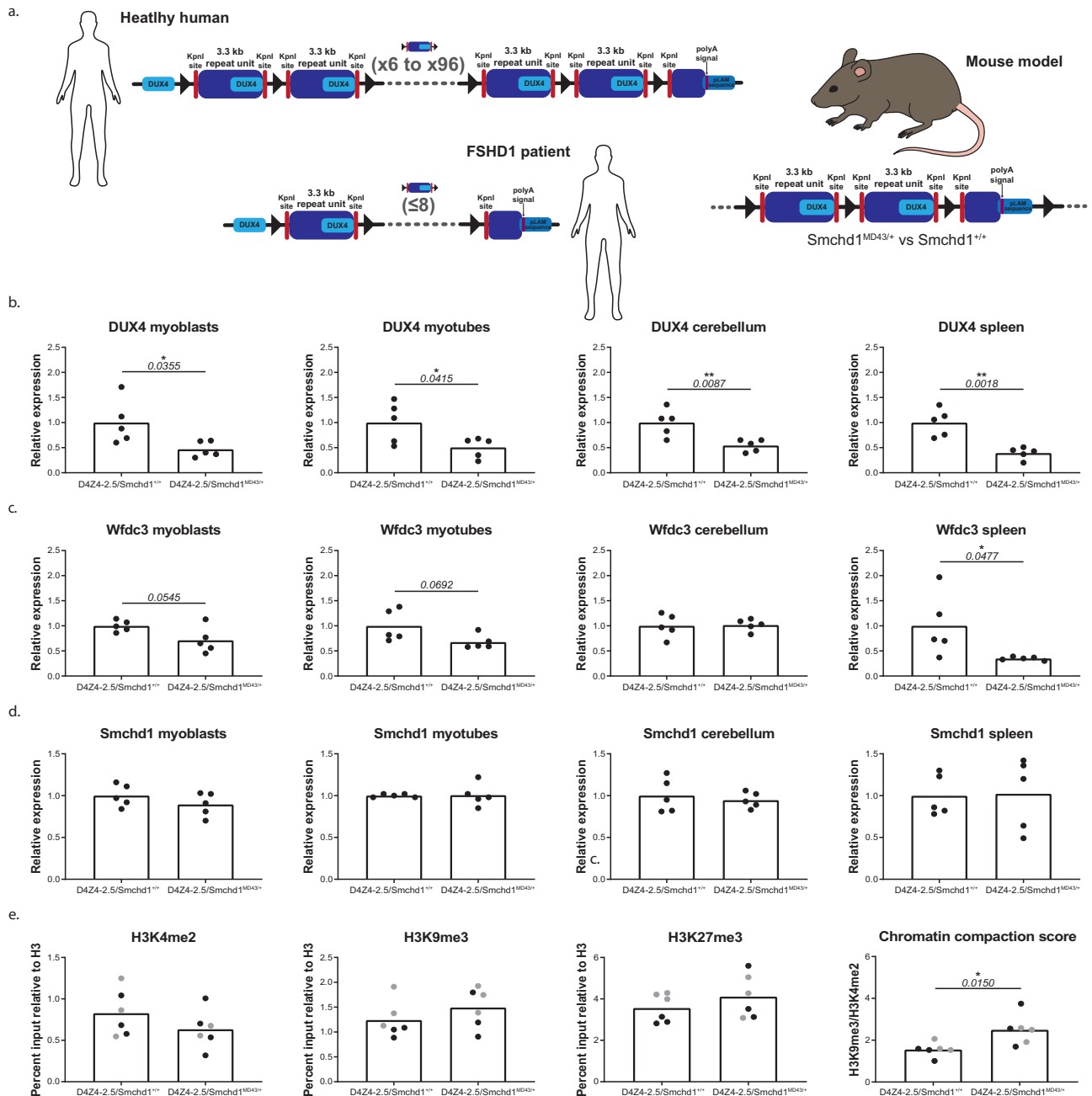

**Fig. 4 | *MommeD43* results in improved silencing of DUX4 in a mouse model of FSHD. a** Diagram of *D4Z4* repeat in healthy humans, FSHD1 patients, and the 2.5-unit transgene repeat used in the murine model (described in ref. 44). **b**–**d** Relative *DUX4*, *Wfdc3*, and *Smchd1* transcript levels in myoblasts, myotubes, cerebellum, spleen, and fibroblasts. Bars represent the average transcript levels per genotype (average value in *D4Z4-2.5* tissue is set as 1); each dot represents a single mouse, *n* = 5 per genotype. Statistical analysis was performed using a Student's *t* test.

*P < 0.05; **P < 0.01, two-tailed. **e** H3K4me2 levels, H3K9me3 levels, H3K27me3 levels, and the chromatin compaction score (H3K9me3 level corrected for H3K4me2 level) in fibroblast cultures (black dots) and spleen (gray dots). Bars represent the average levels per genotype; each dot represents a single mouse, *n* = 3 per genotype per tissue/cell type. Statistical analysis was performed using a Student's *t* test. *P < 0.05, two-tailed. *MommeD43* abbreviated to *MD43*.

In *Smchd1* null NSCs—either *Smchd1^MommeD1/MommeD1^* derived from male mice that have never had SMCHD1 or *Smchd1^del/del^* deleted at least 7 days prior from female *Smchd1^fl/fl^* NSCs in culture—we found major changes in gene expression by RNA-sequencing (1520 DEGs FDR < 0.05 in *MommeD1*[13]; 463 DEGs FDR < 0.05 in *Smchd1^del/del^* cells[14]). Conversely, there were no significant changes in female *Smchd1^MommeD43/MommeD43^* NSCs in either strain background (FVB/NJ from the ENU screen and C57BL/6 Smchd1-GFP fusion created with CRISPR-Cas9). The clustered protocadherins on chromosome 18, which are well known to be regulated by SMCHD1[10,11,13], and whose expression is greatly increased

upon SMCHD1 loss, showed interesting changes, albeit not significant. Although barely detectable, their expression is consistently reduced in both the ENU and CRISPR-induced *Smchd1^MommeD43/MommeD43^* cells, supporting the notion that *MommeD43* is a gain-of-function mutation (Supplementary Fig. 9).

Another known role of SMCHD1 is in mediating long-range chromatin interactions. We performed in situ chromosome conformation capture genome-wide (HiC) in female *Smchd1^GFP/GFP^* and *Smchd1^MommeD43-GFP/MommeD43-GFP^* NSCs (*n* = 3 each). We have previously shown that *Smchd1* deletion caused major changes in chromatin

conformation without altering topologically associated domains (TAD) borders or the distribution of A and B compartments[14]. Similarly, *MommeD43* did not shift TAD borders (Supplementary Fig. 10) or compartments, with the exception of modest differences on the X chromosome (Fig. 5a). The modest changes in chromatin architecture we observed in *MommeD43* NSCs were highly correlated to those seen in *Smchd1*$^{del/del}$ cells ($r = 0.64$ at 1 Mb resolution, Fig. 5b, c, $r = 0.60$ at 100 kb resolution, Fig. 5b, c), including those on the X chromosome. Of the 61 significantly different interactions (FDR < 0.1, 1 Mb resolution, Supplementary Data 5), 50.8% are on the X chromosome (all strengthened in *Smchd1*$^{MommeD43/MommeD43}$), consistent with what was observed in *Smchd1*$^{del/del}$ female cells and SMCHD1's binding across the inactive X chromosome.

Many changes in the *Smchd1*$^{del/del}$ cells were observed between the *Hoxb* cluster and other clustered gene families on chromosome 11 (Fig. 5d). The most significantly weakened of these interactions was between the *Hoxb* and the keratin gene clusters, which was also noticeably weakened in *Smchd1*$^{MommeD43/MommeD43}$ cells (Fig. 5e, f). We focused on a specific interaction between the *Hoxb* cluster and an olfactory receptor gene cluster approximately 50 Mb away, which was significantly weakened upon *Smchd1* deletion in our HiC data, by using DNA Fluorescence in situ hybridization (DNA FISH, Fig. 5g). The results were consistent with our HiC data, showing a significant decrease in the frequency of interaction in *Smchd1*$^{del/del}$ cells (80% decrease, Fig. 5h), and a less pronounced though still significant decrease in *Smchd1*$^{MommeD43/MommeD43}$ cells (ENU and CRISPR-induced mutants, 63% decrease, Fig. 5i). That same interaction was not significantly weakened in the HiC data in *Smchd1*$^{MommeD43/MommeD43}$ cells, which might be due to HiC being a high-background and low-resolution technique less suited to detecting subtle changes. From these data, *MommeD43* seems to behave as a hypomorphic allele with respect to chromatin conformation, which is in contrast to the gain-of-function effect we observed by all other measures of SMCHD1 function.

To further investigate the effects of the *MommeD43* variant on chromatin conformation in a tissue more relevant to the observed phenotypes, and at higher resolution, we performed Capture-C. We used capture probes across SMCHD1 targets (all four *Hox* clusters, the clustered protocadherins, and olfactory receptor clusters) in 7–9 somite PSM tissue from matched wild-type and *MommeD43* mutant samples from the ENU-induced and CRISPR alleles ($n = 2$–$3$ per genotype from a pool of 2 embryos each). As expected based on prior studies from others, we observed two distinct domains within each cluster[54,55]. One shows almost exclusively self-interactions and contains the more posterior *Hox* genes that have not been activated yet. The other contains active genes and is characterized by interactions with other domains outside the cluster (Fig. 5j, Supplementary Fig. 11). With the depth of coverage possible from such low input samples, relatively few of the interactions were statistically significant (CHiCANE[56], Supplementary Data 6), and we did not observe any striking differences in the comparatively short-range interactions within each cluster. We also did not observe major changes in interactions between each cluster and regions outside the captured clusters. While we cannot rule out a subtle or dynamic role in short-range chromatin interactions with these data, taken together with the DNA FISH and in situ HiC data, our findings suggest that SMCHD1 has its most profound role in regulating long-range chromatin interactions. These data are consistent with the role of SMCHD1 in regulating gene expression being separate from its role in chromatin architecture.

Other previously described effects attributable to loss of SMCHD1 during development are DNA hypomethylation at its autosomal targets and on the inactive X chromosome[10,11,14] increased levels and spreading of H3K27me3 on the inactive X chromosome[14]. To examine DNA methylation, we performed Reduced Representation Bisulphite Sequencing (RRBS) in female *Smchd1*$^{MommeD43/MommeD43}$ NSCs. We observed no differences in CpG island methylation levels (only 356

hypermethylated and 81 hypomethylated individual CpG nucleotides found with at least 25% difference $q < 0.05$), either at autosomal regions or on the X chromosome. Similarly, H3K27me3 ChIP-seq in NSCs showed no change in peak localization (Fig. 6a, b, Supplementary Data 7); however, this technique is ill-suited to measure relative levels of histone marks. We turned to immunofluorescence staining to assess relative levels of both SMCHD1 and H3K27me3 in both *Smchd1*$^{del/del}$ and *Smchd1*$^{MommeD43/MommeD43}$ female NSCs, in which we could examine H3K27me3 enrichment specifically on the inactive X chromosome. The X chromosome bears most of the significant alterations in chromatin interactions and shifts in compartment type observed in Hi-C data upon *Smchd1* perturbation (Fig. 5a). After acquiring high-resolution confocal Z-stack images, we defined the nuclear and inactive X volumes by using the boundaries of DAPI (DNA dye) and high H3K27me3 signal respectively, then used the total fluorescence intensity within each channel normalized to the volume of the region as a proxy for relative levels of SMCHD1 and H3K27me3 on the inactive X (Fig. 6d). *Smchd1*$^{del/del}$ cells showed the expected increase in H3K27me3 levels, whereas *Smchd1*$^{MommeD43/MommeD43}$ cells from both the ENU-induced and CRISPR-induced backgrounds showed a modest but significant decrease in H3K27me3 levels, both on the inactive X (Fig. 6e, f) and for the whole nucleus (data not shown). Interestingly, there was a slight, yet significant, reduction in SMCHD1 levels in *Smchd1*$^{MommeD43/MommeD43}$ nuclei (Fig. 6g) that may have been outside the detection range of the western blots and flow cytometry shown previously. Even with slightly decreased SMCHD1 levels the effect observed with the *MommeD43* variant is opposite to that of loss of SMCHD1, which suggests that the gain-of-function effect seen here outweighs any minor change in protein levels.

To further our analysis of the chromatin, we performed ChIP-seq for CTCF in *Smchd1*$^{MommeD43-GFP/MommeD43-GFP}$ (CRISPR allele on C57BL/6 J), *Smchd1*$^{del/del}$ and *Smchd1*$^{+/+}$ female NSCs for comparison. Interestingly, although CTCF binding had minor changes induced by the *MommeD43* mutation (143 differential peaks FDR 0.1, Supplementary Data 8), peaks that had significantly increased binding in the null samples tended to be more weakly bound in the *MommeD43* samples (Fig. 6c, Supplementary Fig. 12). These peaks were not enriched on the X chromosome, as expected given that CTCF is predominantly bound on the active X[36,37] whereas SMCHD1 is on the inactive X. These data suggest *MommeD43* has a gain-of-function effect for both H3K27me3 enrichment on the Xi and CTCF binding on autosomes.

We have shown here that *MommeD43* has either no effect or behaves as a gain-of-function SMCHD1 mutation in a context-dependent manner, except in chromatin conformation where it is more akin to a hypomorphic mutation. Therefore, we believe *MommeD43* is a neomorphic mutation with altered function.

## Discussion

We and others have previously studied SMCHD1 extensively in X chromosome inactivation, development, its effects on gene regulation as well as its role in organizing the genome. It has been shown that SMCHD1 plays a role in *Hox* gene regulation[14], clustered protocadherin gene expression[10,11,13], in mediating long-range chromatin interactions[14,35–37], and in canonical and non-canonical imprinting[12]. Moreover, variants in SMCHD1 that have differing outcomes on SMCHD1 function are associated with two different diseases: FSHD and BAMS[16,31]. In each case, SMCHD1 has been considered an epigenetic repressor, however, a molecular mechanism of SMCHD1-mediated silencing has yet to be elucidated. In this study, we report a new single amino acid substitution mouse variant of SMCHD1, *MommeD43*. This variant encodes the A667E SMCHD1 mutation, which leads to a substitution in the C-terminal portion of the extended ATPase domain. By studying the consequence of this variant on SMCHD1 molecular function and in models of BAMS and FSHD, we enhance both the understanding of how SMCHD1 brings

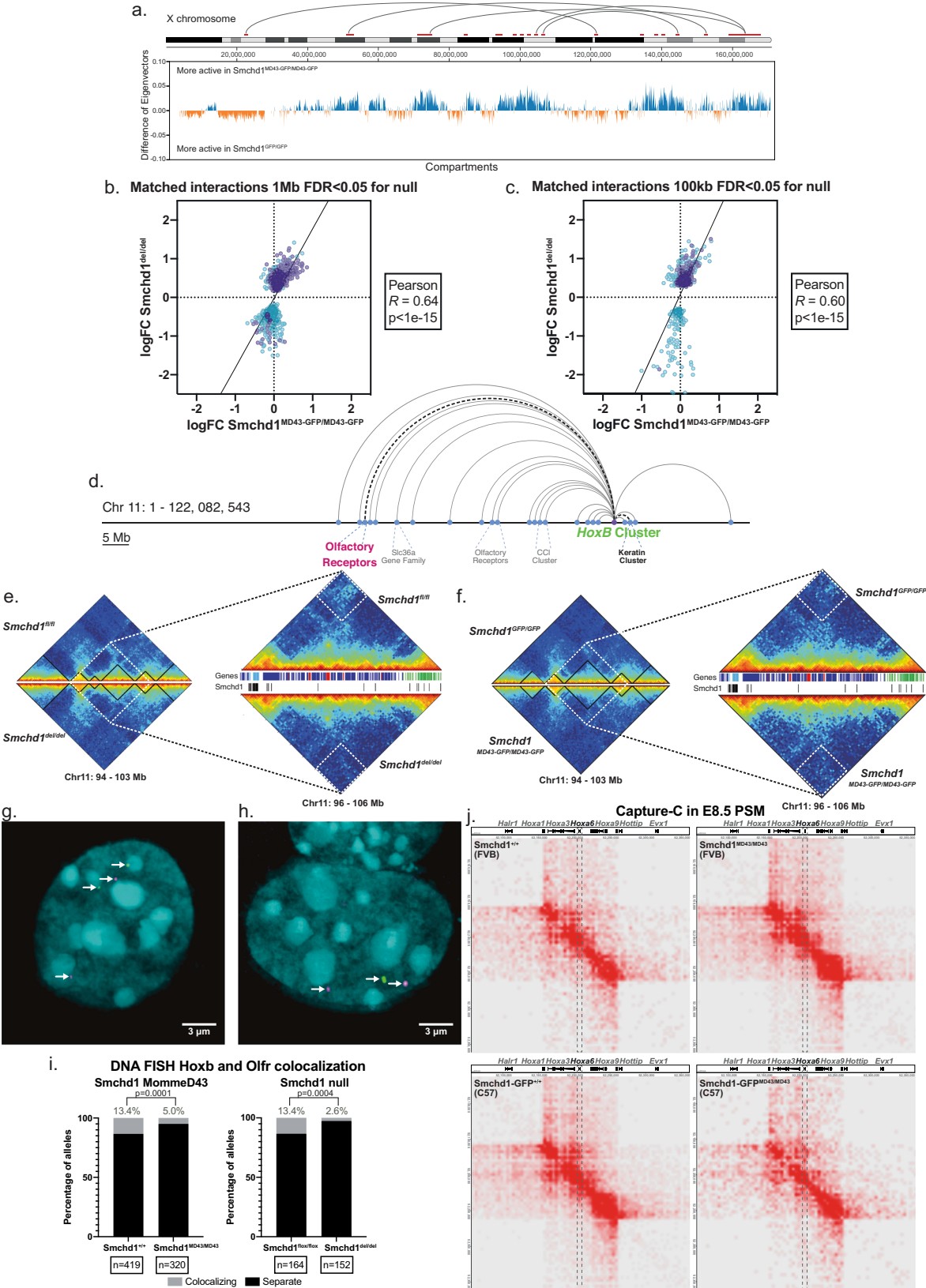

**b. Matched interactions 1Mb FDR<0.05 for null**

**c. Matched interactions 100kb FDR<0.05 for null**

**Capture-C in E8.5 PSM**

**DNA FISH Hoxb and Olfr colocalization**

about gene silencing and identify a potential avenue for treatment of FSHD.

Through our developmental and molecular analyses, we show that, with the level of resolution of the techniques used, there is a counter-intuitive divorce of SMCHD1's activity on gene expression and insulation of the chromatin, versus its effect on chromatin architecture. The *MommeD43* variant was identified because it caused increased transgene silencing. Consistently, we revealed that while its effect on target gene activity is context-dependent, the *MommeD43* variant frequently imparts a gain-of-function effect on SMCHD1. We observed downregulation, which we believe is a delay in activation, of *Hoxa6* and *Hoxd10* in the *MommeD43* mutants. This is consistent with

**Fig. 5 | *MommeD43* has a hypomorphic effect on Smchd1-dependent chromatin interactions. a** Upper panel, X chromosome (IGV) with significant (FDR < 0.1) differential interactions in *Smchd1$^{MommeD43\text{-}GFP/MommeD43\text{-}GFP}$* vs *Smchd1$^{GFP/GFP}$* at 1 Mb resolution (red, strengthened, *n* = 3 per genotype). Gray semi-circles represent the top 5 interactions by fold change. Lower panel, difference in Eigenvectors used to determine A/B compartments between *Smchd1$^{MommeD43\text{-}GFP/MommeD43\text{-}GFP}$* and *Smchd1$^{GFP/GFP}$* (100 kb resolution). **b, c** Differential interactions caused by *Smchd1* deletion between the *Smchd1$^{del/del}$* and *Smchd1$^{MommeD43\text{-}GFP/MommeD43\text{-}GFP}$* at 1 Mb (**b**) and 100 kb (**c**) resolution. Each point represents one interaction, X-linked interactions in purple, two-tailed *p* value. **d** Main SMCHD1-dependent interactions on chromosome 11. **e, f** Left panels, heatmaps of normalized Hi-C interactions in each genotype of female NSCs surrounding the *Hoxb* cluster at 100 kb resolution. Right panels, heatmaps of the region 96–106 Mb region at 50 kb resolution. White dotted squares indicate the most significant interaction lost upon *Smchd1* deletion, also reduced in

*Smchd1$^{MommeD43\text{-}GFP/MommeD43\text{-}GFP}$* cells. The tracks between the heatmaps show the genes (blue−sense, red−antisense, light blue−*Hoxb* genes, green − *Keratin* genes) and previously published Smchd1 ChIP-seq peaks (black). **g, h** DNA FISH with probes labeling the *Hoxb* cluster (green) and the olfactory receptor gene cluster (magenta) highlighted in **a.** in the same colors (DAPI DNA stain, cyan). Left panel, non-interacting loci for both alleles (**g**). Right panel, one non-interacting, and one interacting allele (**h**). **i** Scoring of DNA FISH in g. in male and female NSCs of both the ENU mutant (FVB/NJ) and CRISPR mutant line (C57BL/6 J). Aggregated data from six independent experiments. Two-sided paired *t* test. *MommeD43* abbreviated to *MD43*. **j** Heatmaps of Capture-C interactions centered on the *Hoxa* cluster (*mm10* chr6:52055505-52361195) in presomitic mesoderm tissue from somite-matched embryos (E8.5). A genome browser track shows some of the genes in the region. The dotted line encompasses interactions involving the *Hoxa6* locus, which showed altered expression in this tissue.

the anterior homeotic transformation observed in skeletal patterning, and in contrast to the upregulation of posterior *Hox* genes and posterior homeotic transformation in *Smchd1* null embryos[14]. On the inactive X chromosome, we observed a decrease in H3K27me3 enrichment in the *MommeD43* mutant female cells, whereas the *Smchd1* deleted samples show an increase in H3K27me3. These data suggest that SMCHD1's role in insulating against other epigenetic regulators such as PRC2 and CTCF, as proposed by us and others is enhanced in the *MommeD43* variant[13,14,36,37]. A recent report suggests that SMCHD1 directly interacts with the PRC2 inhibitory factor EZHIP, so we cannot exclude that this effect on H3K27me3 could be due to alteration in this interaction[57]. While these observations suggested a gain of function effect, the influence of the *MommeD43* variant on chromatin interactions were, however, reminiscent of those found upon loss of SMCHD1. We observed weakened long-range interactions between SMCHD1 targets, but to a lesser extent than is observed in the *Smchd1*-null mouse. Our findings suggest that the genome-wide long-range chromatin interactions that require SMCHD1 are not directly linked to SMCHD1's silencing of gene expression. This is consistent with our previous study showing loss of long-range interactions on the inactive X and at *Hox* genes in neural stem cells post *Smchd1* deletion, but no upregulation of these genes when *Smchd1* is deleted after early embryonic development[14].

Many groups have shown the relatively modest effect of altered chromatin interactions on transcription. However, those studies usually rely on the ablation of well-characterized members of the cohesin complex or CTCF (e.g., ref. [58]), which seem to be predominantly architectural in their function. In contrast, newer findings show that PRC1, a complex that has long been considered a direct repressor of gene expression, also plays a role in setting long-range chromatin interactions which is independent of its silencing function[59]. SMCHD1 seems to exert similarly independent effects.

In addition to the *MommeD43* effect on gene silencing, we were intrigued by the depletion of H3K27me3 on the inactive X despite no detectable change in gene expression in our RNA-seq data in NSCs. These data, along with the observed depletion of CTCF binding, suggest that the *MommeD43* variant enhanced SMCHD1's capacity to insulate the chromatin against other epigenetic regulators. Given the loss of long-range interactions on the X chromosome observed in the same cells, SMCHD1-mediated long-range interactions themselves may not offer insulation against other chromatin regulators, but instead must arise from another action of SMCHD1.

We studied the effects of the *MommeD43* mutation on SMCHD1 function related to disease for two main reasons: it is a variant that conferred greater transgene silencing capabilities to SMCHD1 where the transgene array was reminiscent of the *D4Z4* tandem repeat involved in FSHD; and because this enhanced silencing capacity may inform interpretation of SMCHD1 variants found in BAMS, the other

disease where SMCHD1 has a confirmed involvement. Relevant to BAMS, we observed no craniofacial abnormalities in the mouse or *Xenopus* systems; however, we did observe differential expression of some developmentally regulated genes in the FNP regions from *MommeD43* embryos, suggesting that the relevant tissue in the embryo is sensitive to the *MommeD43* variant. For reasons we do not yet understand, these changes did not result in phenotypic outcomes, meaning the MommeD43 mouse is not a useful model for studying mechanisms underlying BAMS.

The gain-of-function effects of the *MommeD43* variant could provide a potentially elegant solution for the treatment of the FSHD phenotype by revealing how to stimulate the wild-type copy of SMCHD1 that is retained in FSHD patients to silence *DUX4*. Indeed, overexpression of SMCHD1, or restoring expression of wild-type SMCHD1 in muscle cell cultures of FSHD2 patients is an efficient way of silencing DUX4[52,60]. Utilizing a mouse model harboring a short *D4Z4* transgene array, we found that the heterozygous *MommeD43* variant in SMCHD1 partially rescues the aberrant activation of *DUX4* in tissues relevant to FSHD. This finding raises the possibility that we could mimic the gain-of-function effect of the *MommeD43* variant therapeutically to re-establish the silencing of *DUX4* in FSHD patients. At this stage, it is not clear how this variant increases SMCHD1's silencing capacity, though our data suggest this may be via an insulation-based mechanism. The activity of SMCHD1 which may be most tractable is its ATPase activity. The precise role of ATPase activity in the execution of SMCHD1's gene-silencing functions is currently incompletely understood. It remains of outstanding interest to define whether this enzymatic activity could be targeted therapeutically, and which functions beyond this enzymatic activity are conducted by the extended ATPase domain, including the C-terminal region where the *Momme43* substitution resides.

## Limitations of the study
In this study we used lines of mice with an ENU-induced *MommeD43* mutation on the FVB/N background, along with the same mutation introduced by CRISPR on the C57BL/6 background. While we replicated our data with both strains for key findings related to long-range interactions, insulation, and gene expression (Supplementary Data 7), it was not feasible to replicate all the embryology and genomics in both lines due to the differing vigor of each strain background and prohibitive cost. Furthermore, in our embryo studies, we have not yet been able to perform an extensive time course throughout development for the short-range interactions by Capture-C, meaning our interpretation is limited to a single timepoint. It is worth noting that the human equivalent of the *MommeD43* mutation has not been observed in BAMS patients, so while we do not observe BAMS-related phenotypes, we have not modeled BAMS-specific mutations in this study. Our work on the *D4Z4 2.5* mouse model also has limitations as this is the human *D4Z4* array placed into the mouse context, where we know muscular

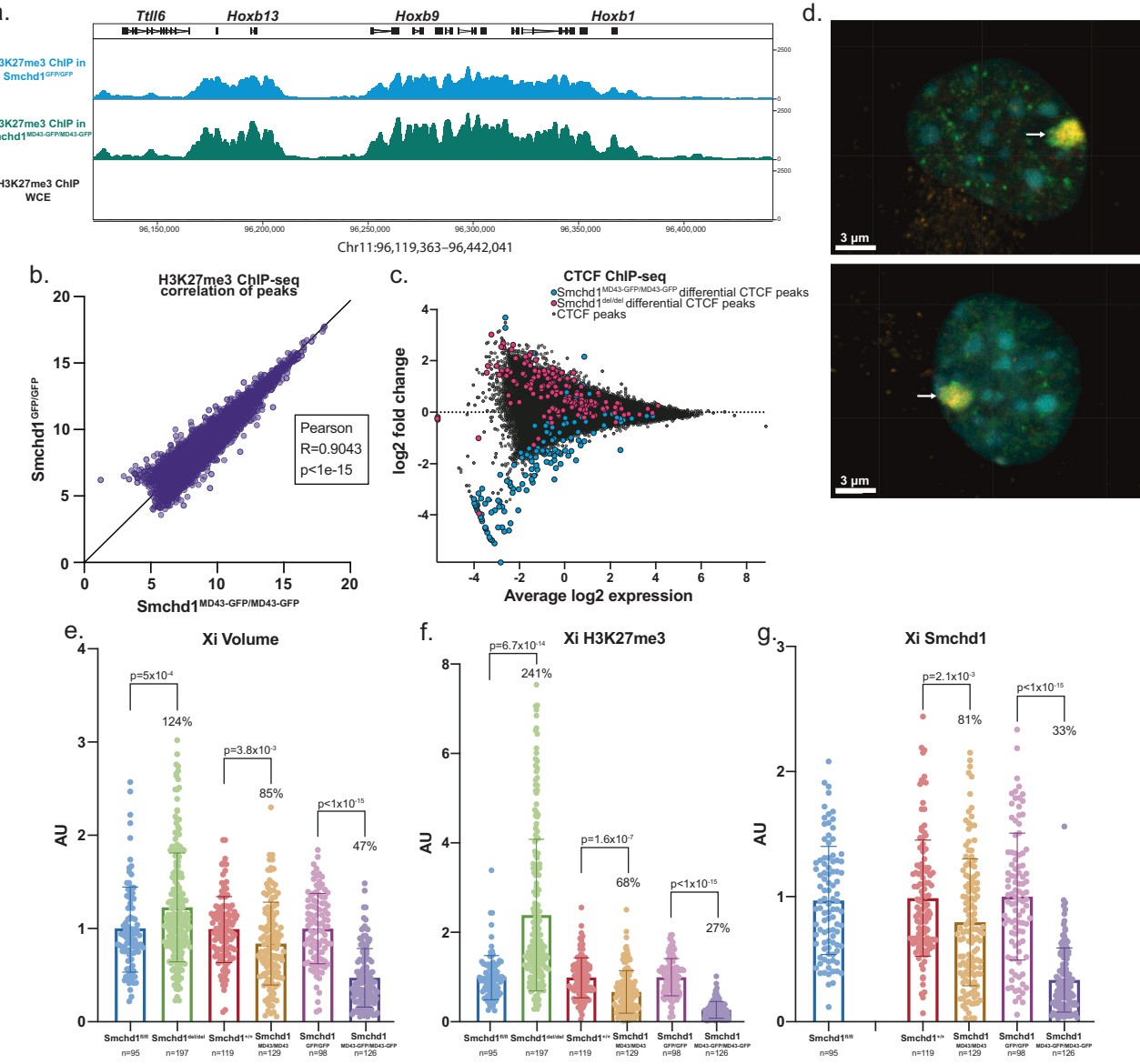

**Fig. 6 | MommeD43 results in depleted H3K27me3 on the inactive X chromosome. a** ChIP-seq track of H3K27me3 in *Smchd1^MommeD43-GFP/MommeD43-GFP* vs *Smchd1^GFP/GFP* primary female NSCs at the *Hoxb* cluster, *n* = 3 biological replicates per genotype. The track shows no differences in H3K27me3 localization (ChIP-seq is not a directly quantitative technique, and so the difference in peak height between the two tracks might not be of biological significance). **b** Scatter plot of log-transformed normalized H3K27me3 ChIP-seq counts in *Smchd1^MommeD43-GFP/MommeD43-GFP* vs *Smchd1^GFP/GFP* NSCs around (±2.5 kb, then merged if within 1 kb) peaks determined from both datasets. The Pearson coefficient indicates very high positive correlation. **c** MA plot showing normalized CTCF ChIP-seq values over peaks. On the *x* axis, *Smchd1^MommeD43-GFP/MommeD43-GFP* - *Smchd1^GFP/GFP* log2 fold change, and on the *y* axis the average between them. Highlighted are peaks showing statistically significant differential binding, in cyan for *Smchd1^MommeD43-GFP/MommeD43-GFP*, and in magenta those found in *Smchd1^del/del*, both compared to *Smchd1^GFP/GFP* (FDR 0.1), *n* = 3 biological replicates per genotype. **d** Representative images of three independent immunofluorescence staining experiments with anti-H3K27me3 (green), anti-SMCHD1

(orange), and DAPI (DNA stain, cyan) in primary female *Smchd1^+/+* (top) and *Smchd1^MommeD43/MommeD43* (bottom) NSCs. The arrow points to the inactive X chromosome which is characterized by very high levels of both H3K27me3 and SMCHD1. **e–g** Scoring (mean ± SD) of the volume and total levels of H3K27me3 and Smchd1 of the inactive X chromosome in female *Smchd1^fl/fl*, *Smchd1^del/del*, *Smchd1^+/+*, and *Smchd1^MommeD43/MommeD43* (ENU mutant and CRISPR mutant) primary female NSCs. Levels of H3K27me3 and Smchd1 are defined as the total intensity of fluorescence inside the volume of the inactive X, normalized by the total nuclear volume of each scored cell, then by the mean of all datapoints in its respective control. The inactive X chromosome is defined as the high H3K27me3 region in a semi-automated approach using Imaris to define a closed surface of highest green fluorescence in each nucleus. *MommeD43* abbreviated to *MD43*. The percentage displayed is the ratio between each *Smchd1* variant and its respective control. *N* provided for each sample in the figure. Unpaired *t* tests two-tailed *p* values displayed for each comparison.

dystrophy does not result, meaning this model is best interpreted in terms of the epigenetic regulation of *D4Z4*. In the future, it may be revealing to study the dynamic role of SMCHD1 at the chromatin to enhance our understanding of function beyond the snapshots provided here.

In conclusion, our study of a new mouse variant of SMCHD1 has disentangled SMCHD1's role in chromatin architecture from its role in transcription and insulation, providing greater mechanistic insight into how SMCHD1 may bring about gene silencing. Moreover, we have revealed that modulating SMCHD1's activity is effective in ameliorating

failed *D4Z4* silencing in a model of FSHD, with direct relevance to FSHD treatment.

## Methods

### Mouse strains and genotyping

All mice were bred and maintained with standard animal husbandry procedures. At WEHI protocols were approved by the WEHI Animal Ethics Committee under animal ethics numbers AEC 2014.026, 2018.004. Mice at QIMR were kept under the approval of the QIMR ethics committee, including the ENU treatment of mice. The approval numbers for this work were QIMR AEC-P1076 and AEC-P1224. At the animal facility of the Leiden University Medical Center all mouse breeding and experiments were performed according to Dutch law and Leiden University guidelines and were approved by the National and Local Animal Experiments Committees. Mice at all three locations were kept in individually ventilated cages with standard rodent chow and water available ad libitum with a standard 12 h/12 h light/dark cycle, 22 degrees Celsius and ambient humidity.

*Smchd1*<sup>fl/fl</sup> mice were maintained on a C57BL/6 background[46]. Mice carrying the *Smchd1*<sup>MommeD1</sup> mutation, as previously described[7], were maintained on the FVB/NJ inbred background. *Smchd1*<sup>GFP/GFP</sup> mice were maintained on a C57BL/6 background[14].

Mice carrying the *Smchd1*<sup>MommeD43</sup> mutation were produced on the FVB/NJ background homozygous for the *Line3* transgene, exactly as previously described for other Momme mutations[7,38,61]. The MommeD43 mutation was backcrossed off the transgenic background and maintained on pure FVB/NJ after mapping of the mutation.

The *MommeD43* mutation was recreated on the *Smchd1*<sup>GFP/GFP</sup> background[14] by injection of Cas9 protein, guide RNA plus repair template into zygotes (sequences for Smchd1 gRNA and repair template in Supplementary Table 3), similarly to what has previously been described[62]. At the same time as introducing the MommeD43 mutation the PAM site was mutated with a silent C to T mutation. This was performed by the WEHI MAGEC facility. The new allele was backcrossed to *Smchd1*<sup>GFP/GFP</sup> for more than 5 generations before animals were used in experiments. This line was created and maintained on a C57BL/6 background.

*D4Z4-2.5* mice were generated before[44]. Female hemizygous *D4Z4-2.5* mice on a C57BL6/J background were crossbred with male heterozygous *Smchd1*<sup>MommeD43/+</sup> mice on an FVB/NJ background to obtain *D4Z4-2.5/Smchd1*<sup>MommeD43/+</sup> mice. Genotypes were determined by PCR analysis using tail DNA.

Both the *MommeD43* and *MommeD1* mutations were genotyped by allelic discrimination (primers in Supplementary Table 3). The *Smchd1*<sup>fl</sup>, *Smchd1*<sup>del</sup> and *Smchd1*<sup>GFP</sup> alleles were genotyped by PCR as previously described[14,46] (primers in Supplementary Table 3). The *D4Z4-2.5* transgene was genotyped as previously described[46] (Supplementary Table 3 for primer sequences).

Embryo tails or yolk sacs were used to prepare DNA through standard methods. Genotypes were determined by PCR with GoTaq Green (Promega) or allelic discrimination for *MommeD1* or *MommeD43*. Sex was determined by PCR for *Otc* (X chromosome) and *Zfy* (Y chromosome) (Supplementary Table 3 for primer sequences).

### ENU mutagenesis screen to identify *MommeD43* mutant line

The ENU mutagenesis screen used to identify the *MommeD43* mutant line of mice was run exactly as previously described[7,38]. In brief, homozygous transgenic males were treated with ENU and allowed to recover fertility. They were bred with wild-type transgenic females and the expression of transgene silencing was assessed in the offspring at weaning, by performing flow cytometry for GFP expression, in a drop of whole blood taken from the tail tip. The founder animal for the *MommeD43* line displayed reduced GFP expression and was bred further to study inheritance patterns.

### Genetic mapping of the *MommeD43* mutation

Mapping of the *MommeD43* mutation was performed as has been reported for other Momme mutations[38]. The *MommeD43* allele was bred down at least 4 generations before mapping was initiated. For mapping the allele was crossed from the FVB/N background to the C57BL/6 background (Line3C, carrying the GFP transgene on the C57BL/6 background) twice to enable positional cloning. These animals were phenotyped for the effect on transgene expression by flow cytometry. The Illumina GoldenGate genotyping assay (Mouse Medium Density Linkage Panel) was performed on 11 phenotypically mutant and 11 phenotypically wild-type littermates. The Mouse Medium Density Linkage panel has 766 measurable SNPs between C57BL/6 J and FVB/NJ. Samples were genotyped following the Illumina protocol and genotype calls were made using the Genotyping module of the GenomeStudio v1.1 software. Only samples with a call rate >95 were accepted. Linked intervals were identified based on LOD score (Supplementary Data 9). A LOD score of over 4.5 was found for chromosome 17, 63–98 Mb. Subsequent exome sequencing using Roche Nimblegen capture in *MommeD43* and wild-type controls revealed the G to T mutation at chr17: 71776840, which relates to a C to A mutation in the sense orientation of exon 15 of *Smchd1*. This mutation was verified by Sanger sequencing. No other mutations were found either within the linked interval or in other exons.

### Nanopore sequencing

Genomic resequencing of *Smchd1*<sup>MommeD43</sup> mice was performed by whole-genome nanopore sequencing. Olfactory bulbs from adult littermate *Smchd1*<sup>+/+</sup> and *Smchd1*<sup>MommeD43/MommeD43</sup> as well as a second *Smchd1*<sup>MommeD43/MommeD43</sup> mouse were dissected and 10 mg processed with the NEB Monarch Genomic DNA Purification kit (T3010) according to manufacturer's instructions. 2 µg input DNA was used for Oxford Nanopore ligation library preparation (SQK-LSK114) according to manufacturer's instructions. 250 ng of the final library were loaded onto PromethION R10.4.1 flow cells (FLO-PRO114M, 4 in total) at 400 bases/s with multiple washes and reloads to achieve 20× coverage per mouse.

Fast5s were basecalled with guppy 6.3.4 in super-accuracy (model dna_r10.4.1_e8.2_400bps_modbases_5mc_cg_sup_prom.cfg) and mapped to mm39 genome with minimap2 in the same step. The epi2me-labs human variation workflow v0.1.1 (https://github.com/epi2me-labs/wf-human-variation) git commit 4497a3b1ad84591-ce28a505959b90c5f4fe4d2ae) was used to call SNVs (with Clair3 citation model r1041_e82_400bps_sup_g615_model) and SVs (with Sniffles2 v2.0.3[63]) in each sample. MommeD43-specific variants were obtained from the intersection of variants in the two *MommeD43* mice that were not in the wild-type control, using bcftools compare v1.16. Variants were then annotated with Ensembl's Variant Effect Predictor tool which also predicts regulatory elements[64]. Variants in the mapped genomic interval (chr17:63 Mb-telomere) were then manually inspected for predicted effect and support using IGV.

### Generation of neural stem cells

NSCs were derived and cultured exactly as previously described[13,14]. In brief, cortices of E14.5 embryos were dissociated and grown as an adherent monolayer on tissue culture-treated plates. Growth media was NeuroCult NSC Basal Medium (Mouse) containing NeuroCult Proliferation Supplement (Mouse) (StemCell Technologies), recombinant human EGF (20 ng/mL), recombinant human basic FGF (20 ng/mL, all from Stem Cell Technologies) and 10 ng/ml laminin (Sigma Aldrich). Primary cells were maintained for a maximum of 20 passages.

### Generation of *Smchd1*<sup>del/del</sup> NSCs

*Smchd1*<sup>del/del</sup> NSCs were generated from *Smchd1*<sup>fl/fl</sup> NSCs exactly as previously described[14]. *Smchd1* deletion was confirmed by immunofluorescence with an in-house anti-Smchd1 antibody[34] during the

immunofluorescence experiments (now available from Merck Cat. No. MABS2292).

## Retrovirus production and transduction

VSVg pseudotyped MSCV-Cre-puromycin retroviral supernatants were produced with calcium phosphate–mediated transient transfection of 293 T cells, as previously described[65]. The medium was collected at 48 h after transfection, centrifuged to remove residual 293 T cells and either concentrated with PEG or used unconcentrated.

Transduction of NSCs with PEG-concentrated viral supernatant was performed exactly as previously described[14].

## Immunofluorescence

Immunofluorescence was performed on $Smchd1^{+/+}$, $Smchd1^{MommeD43/MommeD43}$, $Smchd1^{fl/fl}$ and $Smchd1^{del/del}$ early passage primary NSCs[66]. Plated cells were washed in PBS before fixation for 10 min in 3% (w/v) paraformaldehyde at room temperature. Cells were washed thrice for 5 min in PBS, blocked for 15 min in 1% (w/v) bovine serum albumin (BSA), and then set up with primary antibody (rat anti-Smchd1, in-house monoclonal #8 at 1:100 dilution, rabbit anti-H3K27me3 Cell Signalling technologies, C36B11, 1:100 dilution) for 45 min stain in a humidified chamber at room temperature. After washing thrice in PBS the secondary antibody stain was performed for 40 min in 1% BSA (w/v) in the dark (donkey anti-rabbit-594 Thermofisher Australia Scientific, A-21207, 1:1000 dilution, donkey anti-rat-488 Thermofisher Australia Scientific, A-21208, 1:1000 dilution). After three additional PBS washes, cells were mounted in Vectashield H-1000 mounting medium (Vector Laboratories). Cells were visualized on an LSM 880 (Zeiss) microscope. All images were taken using the same settings and the same instrument across multiple sessions. Images for each "treatment" condition and its corresponding control were acquired within the same sessions. The mean value of the corresponding control was used to normalize total intensity values individually for each channel across different acquisition sessions.

Imaris Software (Bitplane) was used to measure the volume of the nucleus (using DAPI staining) or occupied by high levels of H3K27me3 as marker of the Xi within the nucleus. A threshold was manually set to measure the signal only above nucleoplasmic staining or background. Nuclei that were too close together to be defined as separate volumes were manually removed from subsequent analyses. A region of interest was then defined on the basis of DAPI (nuclei) or focal H3K27me3 enrichment (Xi). The volume was then calculated for the region of interest above the threshold. Total fluorescence intensity levels of Smchd1 or H3K27me3 fluorescence were normalized by the nuclear volume of each cell individually, and then across each acquisition session.

## DNA FISH

DNA FISH was performed on $Smchd1^{MommeD43/MommeD43}$ and its $Smchd1^{+/+}$ control (FVB/NJ background), $Smchd1^{MommeD43-GFP/MommeD43-GFP}$ and its $Smchd1^{GFP/GFP}$ control (C57BL/6 J background) and $Smchd1^{del/del}$ and its $Smchd1^{fl/fl}$ control (C57Bl/6 background) NSCs. 1 μg RP23-196F5 (HoxB locus) or RP24-323I2 (Olfr locus) BAC DNA (CHORI) was used in a 12-hour nick-translation reaction (Vysis) to generate DNA probes labeled with Green 496 dUTP or Orange 552 dUTP (Seebright), respectively. Approximately 100 ng probe per sample was precipitated in ethanol with 10% NaOAc, 1 μg salmon sperm DNA (Life Technologies), and 1 μg mouse Cot-1 DNA before being resuspended in formamide (Sigma Aldrich), denatured at 75 °C for 10 min and allowed to compete with Cot-1 DNA for 1 h at 37 °C. Cells were prepared for DNA FISH[66] by fixing with 3% (w/v) paraformaldehyde for 10 min at room temperature, followed by two washes with PBS for 5 min, then permeabilization in 0.5% Triton X-100 (in PBS) on ice for 5 min. Cells were washed twice in 70% ethanol for 5 min and then dehydrated through a series of

increasing ethanol concentrations (80%, 95%, 100%) for 3 min each. Air-dried samples were then denatured in 50% formamide, 2XSSC (pH 7.2) for 45 min at 80 °C. After washing in ice-cold 2XSSC three times, the probe was hybidised for 36 h of hybridization at 42 °C in the dark in a humidified chamber. Cells were washed thrice in 50% formamide, 2XSSC (pH 7.2) at 42 °C for 5 min each then 2XSSC twice for 5 min before DNA counterstaining with DAPI. Stained cells were then mounted in H1000 Mounting Medium (Vectashield) and visualized on an LSM 880 (Zeiss) microscope with Airyscan processing. Images were analyzed with the open-source ImageJ distribution package FIJI[67]. Brightness and contrast were manually set for each image for clear scoring, and spectral shift was corrected using an image of Tetraspeck 0.1um beads (ThermoFisher) acquired on each session with the same settings as the experiment images. Overlapping or touching FISH signals for HoxB and Olfr probes were scored as interacting. 419 alleles were scored for the Smchd1$^{+/+}$ NSCs, 320 alleles for the Smchd1$^{MommeD43/MommeD43}$ NSCs, 164 alleles for the Smchd1$^{flox/flox}$ and 152 alleles for the Smchd1$^{del/del}$.

## SAXS[26]

SAXS data were collected on the SAXS/WAXS beamline at the Australian Synchrotron, coupled with in-line size exclusion chromatography. 50 μL at 5 mg/mL of wild-type or A667E recombinant SMCHD1 protein (residues 111–702) were sample was each loaded onto a Superdex-200 5/150 (GE Healthcare) pre-equilibrated in purification buffer [200 mM NaCl, 20 mM Tris-HCl (pH 8.0), 10% (v/v) glycerol, 0.5 mM TCEP] and eluted via a 1.5 mm glass capillary at 8 °C positioned in the X-ray beam in which a lamellar sheath of buffer prevents capillary fouling[68,69]. Diffraction data were collected with a 1 M, 170 mm × 170 mm Pilatus detector at 2 s intervals over the course of the elution. Data were processed by the beamline control software, ScatterBrain. 2D intensity plots from the size exclusion chromatography peak of the eluting protein sample were radially averaged and normalized to sample transmission. Scattering profiles from buffer alone were averaged for background subtraction of 1D profiles.

Data analyses were performed with the ATSAS suite[68]. PRIMUS[69] was used to perform Guinier analysis for examining scattering curves at small angles (qRg below 1.3). From this, an estimation of two parameters were obtained: the radius of gyration (Rg) value, which represents the square root of the average distance of each scattering atom from the particle center, and zero angle intensity (I(0)), which is proportional to the molecular weight and the concentration of the protein. The linearity of the Guinier plot reflects the quality of the scattering data obtained, indicating the absence of high molecular weight aggregates or inter-particle interference. Real-space interatomic distance distribution function, P(r), and maximum dimension of the scattering particle, Dmax, were computed by indirect Fourier transform via GNOM[70].

## Protein expression and purification

The SMCHD1 N-terminal region (residues 111–702) was PCR-amplified from a *Mus musculus* full-length Smchd1 template. The *MommeD43* mutation (A667E) was introduced by oligonucleotide-directed mutagenesis (5′ CTGTGCCCATTGAAAAGCTGGAT AGG; 3′ CCTATCCA GCTTTTCAATGGGCACAG) and ligated into the pFastBac Htb vector (Life Technologies). Bacmids were prepared using the Bac-to-Bac system, and utilized for protein expression in *Sf*21 insect cells, as described previously[26]. Cells were maintained in Insect-XPRESS protein-free insect cell media with L-glutamine (Lonza) and infected at a density of 3–4 × 10^6 with high-infectivity baculovirus for protein expression.

Purification was performed as previously described[24,30]. Cells were resuspended in lysis buffer [0.5 M NaCl, 20 mM Tris-Hcl (pH 8.0), 20% (v/v) glycerol, 5 mM imidazole (pH 8.0), 0.5 mM TCEP] supplemented with 1 mM PMSF and 1× cOmplete EDTA-free protease inhibitor (Roche). Sonication was performed on ice for 5 cycles with 1 s on, 0.2 s

off, 22 s per cycle at 50% amplitude, using the Bandelin sonicator fitted with the VS 70/T probe. To remove insoluble material, lysates were centrifuged at $45,000 \times g$ for 30 min at 4 °C. Following cell lysis by sonication, lysate supernatant from cells expressing N-terminal His$_6$-tagged proteins were incubated with nickel-nitrilotriacetic acid (Ni-NTA) cOmplete His-tag purification resin (Roche) for 1 h at 4 °C, on rollers. 1 ml of 50% resin slurry was used per 1 L of cell culture. The resin was pelleted by centrifugation at $1500 \times g$ for 5 min at 4 °C and the supernatant was removed as the unbound sample. The resin was washed twice with 5 mM imidazole buffer (pH 8.0), followed by two washes with 35 mM imidazole (pH 8.0) and eluted in 250 mM imidazole buffer (pH 8.0). The 6-His-tag was cleaved by incubation of the pooled elutions with tobacco etch virus (TEV) protease overnight at 4 °C. Next day, cleaved protein was concentrated with a 30-kDa molecular mass cutoff concentrator (Millipore) by centrifugation for 5 min at $2500 \times g$, 4 °C, then diluted in Buffer A [50 mM NaCl, 25 mM HEPES (pH 7.5), 0.5 mM TCEP, 10% (v/v) glycerol] for ion exchange chromatography. The concentrated protein sample was loaded onto a MonoQ 5/50 GL column (GE Healthcare) pre-equilibrated with Buffer A and exchanged into Buffer B [500 mM NaCl, 25 mM HEPES (pH 7.5), 0.5 mM TCEP, 10% (v/v) glycerol] in a 0–100% gradient over 20 column volumes for protein elution. Fractions of interest were pooled, concentrated, and subjected to size exclusion chromatography on a Superdex-200 10/300 GL column (GE Healthcare) pre-equilibrated with SEC100 buffer [100 mM NaCl, 20 mM HEPES (pH 7.5)]. Fractions containing the recombinant protein of interest were pooled concentrated and snap-frozen in liquid nitrogen for storage at −80 °C. Protein concentration was measured using a DS-11 FX Spectrophotometer (DeNovix) from A280.

## ATPase assay

ATPase assays were performed using the procedure of Chen et al.[26] 10 µL reactions were set up in triplicates in 384-well low flange, black, flat-bottom plates (Corning) containing 7 µL reaction buffer [50 mM HEPES (pH 7.5), 4 mM MgCl2, 2 mM EGTA], 1 µL recombinant protein at concentrations ranging from 0.1–0.6 µM or SEC buffer control, 1 µL nuclease-free water and 1.25–10 µM ATP substrate. Reactions were incubated at 25 C for 1 h in the dark. Reactions were stopped by the addition of 10 µL detection mix [1× Detection buffer, 4 nM ADP Alexa-aFluor 633 Tracer, 128 µg/mL ADP$^2$ antibody] and incubated for another hour in the dark. Fluorescence polarization readings (mP) were measured with an Envision plate reader (PerkinElmer Life Sciences) fitted with excitation filter 620/40 nm, emission filters 688/45 nm (s and p channels) and D658/fp688 dual mirror. Readings from a free tracer (no antibody) control were set as 20 mP as the normalization baseline of the assay for all reactions. The amount of ADP produced by each reaction was estimated by a 12-point standard curve, as outlined in the manufacturer's protocol. Data was plotted and analyzed in GraphPad Prism.

## Western blot

Samples were resolved by standard reducing SDS-PAGE analysis on 4–12% Bis-Tris gels (Thermo Fisher Scientific) in MES buffer and transferred to a PVDF membrane (Osmonics, GE Healthcare) by wet transfer at 100 V for 1 h in transfer buffer [25 mM Tris, 192 mM glycine, 20% (v/v) methanol]. Membranes were blocked with a 5% (v/v) skim milk powder in 0.1% (v/v) Tween-20/PBS for 1 h at room temperature. Primary antibody (monoclonal in-house anti-Smchd1, clone 1D6, 1:2000 or anti-Tubulin SantaCruz Biotechnology, SC-23948, 1:5000) was added to the membranes in 5 mL blocking buffer and incubated overnight at 4 °C in a capped tube, on rollers. Membranes were washed for 30 min at room temperature with 0.1% (v/v) Tween-20/PBS, followed by incubation with secondary antibody (anti-rat IgG HRP-conjugated, Southern Biotech, 3030-05, 1:10,000, goat anti-mouse IgG HRP-conjugated, Southern Biotech, 1036-05, 1:10,000) for 1 h at room temperature, which was diluted in 5 mL blocking buffer. The 30 minute washing step was repeated and antibody binding was visualized using the Luminata ECL system (Millipore) following the manufacturer's instructions.

## Skeletal preparations and scoring

Whole-mount skeletal staining of E16.5 fetuses was performed as previously described[71]. Each fetus was skinned and organs were removed under a light microscope, in addition to dissolving remaining tissue in acetone. Once stained, skeletons were washed through a graded glycerol/water series before imaging in 100% glycerol using a ZEISS SV11 stereomicroscope. The vertebral phenotype of each skeleton was scored by two independent assessors who were blind to genotype and sex.

## Tailbud dissection and somite counting

Tailbud tissue containing the presomitic mesoderm (PSM) was dissected from *Smchd1*$^{+/+}$ and *Smchd1*$^{MommeD43/MommeD43}$ E8.5 embryos as previously described[14,72]. In brief, embryos were dissected ice-cold DEPC-treated PBS. Tailbud tissue was horizontally dissected at the distance of 1.5 somites below the last segmented somite to ensure the next developing somite from the PSM, S0, was not included in the tailbud dissection. Tailbud tissue was snap-frozen on dry ice and stored at −80 °C for later RNA extraction using a Zymo Quick-RNA Miniprep Kit. The yolk sac was retained for genotyping. Somites were counted before fixing each embryo in 4% DEPC-treated paraformaldehyde at 4 °C overnight. Embryos were washed through a graded methanol/PBT (DECP-treated PBS with 1% Tween (v/v)) series as previously described[73] before brief staining in dilute ethidium bromide solution and imaging under a fluorescence dissection microscope to check somite counting. RNA-sequencing was performed on somite-matched *Smchd1*$^{+/+}$ and *Smchd1*$^{MommeD43/MommeD43}$ tailbud tissue.

## Frontonasal prominence collection

E10.5 embryos were dissected in ice-cold PBS and the FNP removed by dissecting along the boundary which separates the FNP from the maxillary prominence using fine forceps. Dissected tissue was immediately snap-frozen on dry ice and the remaining embryo was placed in a fresh well of PBS for somite counting. Yolk sacs were collected for genotyping purposes.

## HREM

E14.5 embryos were collected and fixed in Bouin's fixative overnight followed by extensive washing and storage in PBS. Following dehydration in a graded methanol series, samples were incubated for 3 days in JB-4 (Sigma)/Eosin (Sigma)/Acridine orange (Sigma) dye mix before embedding and imaging as previously described[74,75] (https://dmdd.org.uk/hrem/). To measure key craniofacial features, samples were aligned to equivalent orientations in three dimensions, and the distance measured then divided by crown-rump length using OsirixMD. Statistics were calculated using unpaired $t$ tests, followed by Benjamini–Hochberg procedure to correct for the false discovery rate.

## *Xenopus* embryological assays

*Xenopus laevis* was used according to guidelines approved by the Singapore National Advisory Committee on Laboratory Animal Research. Injections of human *SMCHD1* mRNA into Xenopus embryos were performed as previously reported[16]. Briefly, two dorsal animal blastomeres were injected at the 8-cell stage with 240 pg of in vitro transcribed human *SMCHD1* mRNA containing various mutations. Embryos were allowed to develop at room temperature until stage 45–46 and fixed. Eye diameter was measured using a Leica stereomicroscope with a DFC 7000 T camera. No statistical method was used to predetermine sample size. No randomization or blinding was used. Embryos that died before gastrulation were excluded. Statistics were

calculated using one-way ANOVA, followed by Dunn's multiple comparison test.

## Isolation and culturing of fibroblasts and EDL-derived muscle cells

For establishing fibroblast cultures, neonatal mice were sacrificed by decapitation after which the skin was removed. The skin was next cut into small pieces, evenly distributed over a 60 mm culture dish, and a small amount of DMEM/F12 medium (Gibco; Thermo Fisher Scientific, Bleiswijk, the Netherlands) supplemented with 1% penicillin-streptomycin (P/S) (Sigma Aldrich, Zwijndrecht, the Netherlands) and 20% heat-inactivated fetal bovine serum (FBS) (Thermo Fischer Scientific, the Netherlands) was added. Twice a week, a small amount of medium was added to the culture dish, and over time fibroblasts grew out of the skin pieces. These cells were either transferred for further expansion using the same medium or harvested for RNA or chromatin isolation.

For establishing muscle cell cultures, mice of two months of age were sacrificed by cervical dislocation followed by removal of the EDL muscle from tendon to tendon. The EDL muscle was next incubated for 105 min at 37 °C in 0.2% collagenase (Sigma Aldrich, Zwijndrecht, the Netherlands) in DMEM medium (Gibco; Thermo Fisher Scientific, Bleiswijk, the Netherlands) containing 1% P/S (Sigma Aldrich, Zwijndrecht, the Netherlands). The resulting individual muscle fibers were dissociated with a Pasteur pipet with a smooth end and, after several washing steps as previously described[76], transferred to a Matrigel-coated (BD Biosciences, Vianen, the Netherlands) 6-well culture plate (150 fibers per well) in DMEM medium supplemented with 30% FBS, 10% horse serum, 1% P/S, 1% chicken embryo extract, and 2.5 ng/ml fibroblast growth factor (all from Thermo Fischer Scientific, Bleiswijk, the Netherlands). After three days, the fibers were removed, and the attached myoblasts were trypsinised and plated in fresh Matrigel-coated plates. Myoblasts were harvested for RNA at -70% confluency or differentiated by replacing the medium with DMEM with 2% horse serum and 1% P/S. 72 h after the start of differentiation, myotubes were harvested for RNA isolation.

## RNA isolation, cDNA synthesis, and real-time quantitative PCR

Total RNA was isolated using the miRNeasy kit (Qiagen, Venlo, the Netherlands) following the manufacturer's instructions and included a DNase treatment on the column for 30 min at room temperature. RNA concentrations were determined using the Nanodrop ND-1000 spectrophotometer (Thermo Fisher Scientific, Bleiswijk, the Netherlands). 1–3 μg RNA was reverse transcribed with the RevertAid H Minus First Strand cDNA synthesis kit and Oligo(dT)18 primers (both Thermo Fisher Scientific, Bleiswijk, the Netherlands), following the instructions of the manufacturer. cDNA was treated with 2 units of RNaseH (Thermo Fisher Scientific, Bleiswijk, the Netherlands) for 20 min at 37 °C. RT-qPCR analysis was performed with the CFX96 system (Bio-Rad, Veenendaal, the Netherlands) using iQ SYBR Green Supermix (Bio-Rad, Veenendaal, the Netherlands), 0.5 pM of each primer (sequences are listed in Supplementary Table 3), and 1:5 or 1:50 diluted cDNA. The following cycling conditions were used: an initial denaturation step at 95 °C for 3 min followed by 40 cycles of 10 seconds at 95 °C and 30 seconds at primer Tm. A melting curve analysis from 65 °C to 95 °C (temperature increments of 0.5 °C) was performed to determine the specificity of each reaction. Data were analyzed with Bio-Rad CFX Manager version 3.1 (Bio-Rad, Veenendaal, the Netherlands) and normalized to the housekeeping genes Gapdh and Rpl13a.

## LiCor western blot in NSCs and D4Z4 samples

The LiCor Western blot for SMCHD1 in NSCs was performed as described above under western blot with some variations. The primary antibodies were against SMCHD1[14] (monoclonal in-house anti-SMCHD1, clone 1D6, 1:2000) or HSP90 (Abcam, ab13492, 1:1000).

Secondary antibodies used were goat anti-Rat IgG- 680 (LiCor 926–68076, 1:10,000 dilution) and donkey anti-Mouse IgG- 800 (LiCor 926–32212, 1:10,000 dilution Li-Cor, Bad Homburg, Germany). Blots were imaged using the Odyssey CLx imager (Li-Cor, Bad Homburg, Germany).

Tibialis anterior muscle and spleen tissue were homogenized in 10 volumes of solubilization buffer (150 mM NaCl, 50 mM Tris, 200 mM PMSF, 100 mM benzamidine, pH 7.4) with 1% Triton X-100. Next, samples were incubated, rotating top-over-top, for 1 h at 4 °C and centrifuged for 30 min at 4 °C at maximum speed (tabletop centrifuge) to remove non-homogenized material. Protein concentrations were determined using the Pierce™ BCA Protein Assay Kit (Thermo Fisher Scientific, Bleiswijk, the Netherlands). Protein samples were separated on a NuPAGE™ 4–12% Bis-Tris Protein Gel (Thermo Fisher Scientific, Bleiswijk, the Netherlands) and transferred to an Immobilon-FL PVDF membrane (Merck, Amsterdam, the Netherlands). The membrane was blocked for 1 h at room temperature in 4% skim milk/PBS containing 0.1% Tween-20, followed by an overnight incubation step at 4 °C with primary antibodies: anti-Smchd1 antibody (1:250; HPA039441; Sigma Aldrich, Zwijndrecht, the Netherlands) and anti-Emerin (1:200; SC-15378; Santa Cruz Biotechnology; Bio-Connect B.V., Huissen, the Netherlands). The next day, blots were washed with PBS containing 0.1% Tween-20 and incubated with dye-conjugated secondary antibodies (goat anti-rabbit-800, 926–32211, 1:10,000 Li-Cor, Bad Homburg, Germany) for 1 h at room temperature in 4% skim milk/PBS with 0.1% Tween-20. After washing with PBS containing 0.1% Tween-20 and PBS, blots were imaged using the Odyssey CLx imager (Li-Cor, Bad Homburg, Germany).

## DNA methylation analysis at *D4Z4*

Genomic DNA (400 ng) from mouse tail was bisulphite converted with the EZ DNA Methylation-Lightning kit (Zymo Research; BaseClear Lab Products, Leiden, the Netherlands) following the instructions of the manufacturer. A PCR reaction of the DR1 region within the *D4Z4* repeat transgene and a PCR reaction of the FasPas region just distal of the *D4Z4* repeat transgene was performed using FastStart Taq DNA polymerase (Roche, Woerden, the Netherlands) with the following cycling conditions: initial denaturation for 10 min at 95 °C followed by 35 cycles of 20 seconds at 95 °C, 30 seconds at 60 °C and 40 seconds at 72 °C, with a final extension step for 5 min at 72 °C. Next, the PCR products were ligated into the TOPO TA vector (Thermo Fisher Scientific, Bleiswijk, the Netherlands), followed by transformation of the ligation products into competent DH5α bacteria. Plasmid DNA from at least 10 individual colonies was isolated and sent for Sanger sequencing.

## Chromatin immunoprecipitation followed by real-time quantitative PCR at *D4Z4*

We followed a previously published protocol with minor modifications[77]. Confluent cultures of neonatal fibroblasts and spleen tissue minced into small pieces were used for these analyses. Both cells and tissues were crosslinked with 1% formaldehyde for 10 min at room temperature, after which the cross-linking reaction was stopped by adding 125 mM glycine for 5 min at room temperature. Next, fibroblasts were lysed in NP buffer (150 mM NaCl, 50 mM Tris-HCL (pH 7.5), 5 mM EDTA, 0.5% NP-40, 0.1% Triton X-100). Spleen tissue was lysed in LB buffer (50 mM Hepes (pH 7.5), 14 mM NaCl, 1 mM EDTA, 10% glycerol, 0.25% NP-40, 0.25% Triton X-100). The resulting chromatin was sheared in NP buffer using a sonicator bath (Bioruptor Pico; Diagenode, Ougrée, Belgium) for 30 min at maximum output and 30 seconds on/off cycles. The fragmentation between 200–2000 bp was confirmed through phenol-chloroform extraction followed by agarose gel electrophoresis. DNA concentrations were determined with the Nanodrop ND-1000 spectrophotometer (Thermo Fisher Scientific, Bleiswijk, the Netherlands). For the histone ChIP analyses, 3 μg of

chromatin was used. For the Smchd1 ChIP analysis, 30 µg of chromatin was used. First, chromatin was precleared for 1 h at 4 °C with blocked protein A Sepharose beads (GE Healthcare, Eindhoven, the Netherlands). Next, precleared chromatin was incubated overnight at 4 °C with 5 µg of the following antibodies: H3K4me2 (39141; Active Motif, Carlsbad, USA), H3K9me3 (39161; Active Motif, Carlsbad, USA), H3K27me3 (07-449; Merck, Amsterdam, the Netherlands), Smchd1 (ab31865; Abcam, Cambridge, United Kingdom), or mouse IgG (PP64; Merck, Amsterdam, the Netherlands). Immunoprecipitation was performed by incubating the antibody-chromatin mixture with blocked protein A Sepharose beads for 2 h at 4 °C. Washing of the beads was performed according to the previously published protocol[77]. DNA was isolated with Chelex resin (Bio-Rad, Veenendaal, the Netherlands). Finally, quantitative PCR analysis was performed to amplify the D4Z4 repeat transgene[49]. The Gapdh promoter was amplified as a control.

## Statistical analyses

Statistical analyses of the non-genomic data were performed with GraphPad Prism software (version 8; GraphPad Software, Inc., La Jolla, USA). The statistical tests that were performed are described in the figure legends. $P < 0.05$ was considered significant.

**Genomics.** All of the genomics datasets were analyzed from the raw data stage concomitantly with all the other datasets to which they were compared. In the case of ChIP-seq in $Smchd1^{GFP/GFP}$ NSCs, RNA-seq in $Smchd1^{MommeD1/MommeD1}$ or $Smchd1^{fl/fl}$ and $Smchd1^{del/del}$ NSCs, in situ Hi-C in $Smchd1^{fl/fl}$ and $Smchd1^{del/del}$ NSCs, these data were published previously.

## ChIP-seq

Chromatin immunoprecipitation for Smchd1-GFP was performed exactly as described in Wanigasuriya, Gouil et al.[12] in $4 \times 10^7$ cells from three independent primary NSC lines for each genotype $Smchd1^{MommeD43-GFP/MommeD43-GFP}$ and $Smchd1^{GFP/GFP}$. Formaldehyde cross-linking was performed at 1% (v/v) final concentration for 10 min, followed by glycine quenching. Pelleted cells were washed with PBS twice then frozen on dry ice. Pellets were thawed in ChIP buffer (150 mM NaCl, 50 mM Tris-HCl pH 7.5, 5 mM EDTA, 0.5% vol/vol Igepal CA-630, 1% Triton X-100, 1× cOmplete cocktail (Roche)) and dounced 25 times on ice in a tight dounce. Nuclei were pelleted (12,000 × g, 1 min, 4 °C) and washed with ChIP buffer before MNase digestion (NEB) with $2 \times 10^4$ U of MNase for 5 min after a 5 min preincubation in MNase buffer. The MNase reaction was stopped with 10 mM EDTA on ice. After pelleting nuclei again (12,000 g, 1 min, 4 °C), fragmentation was performed in a Covaris S220 sonicator (peak power, 125; duty factor, 10; cycle/burst, 200; duration, 15 s) in Covaris microTubes, 520 µL per tube. After clearing debris by centrifugation (12,000 × g, 1 min, 4 °C), supernatant is kept. A sample was taken for WCE and the remainder used for immunoprecipitation with 16 µg of anti-GFP antibodies (Invitrogen A11122) overnight at 4 °C with rotation. Following this, chromatin was pelleted by centrifugation (12,000 × g, 10 min, 4 °C). The antibody-bound chromatin was then collected by suspending in 80 µL of protein G DynaBeads (ThermoFisher, washed 3 times in cold ChIP buffer right before use) followed by incubating at 4 °C for 1 h with rotation. Beads and chromatin were washed in ice-cold PBS, six times, before eluting chromatin with Elution buffer (1% SDS, 0.1 M NaHCO₃) twice by incubation for 15 min with rotation at room temperature. Overnight incubation at 65 °C with 8 µL of 5 M NaCL and 1 µL of RNase A (NEB) for every 200 µL of eluate or WCE enabled the reversal of cross-links. Samples were digested with proteinase K for 1 h at 65 °C before DNA was extracted with Zymo clean and concentrator kit.

Chromatin immunoprecipitation for H3K27me3 was performed exactly as described in Jansz et al.[14] "ChIP for histone proteins". $1 \times 10^6$ cells from the same cell lines as above were used with 2 µg of anti-H3K27me3 antibody (Cell Signalling Technologies, C36B11). This method is as per the above paragraph except that the nuclei are isolated in 20 mM Tris pH 8.0, 10 mM NaCl, 2 mM EDTA pH 8.0, 0.5% Igepal CA-630, 1× cOmplete protease inhibitor (Roche), and then subject to fragmentation in 20 mM Tris pH 7.5, 150 mM NaCl, 2 mM EDTA, 1 % Igepal CA-630, 0.3% Sodium dodecyl sulfate, 1× cOmplete protease inhibitor (Roche), on the Covaris S220 (peak power: 105, duty factor:20, cycle/burst: 200, duration: 750 s). The sample was diluted for immunoprecipitation with 20 mM Tris-HCl pH 8.0, 150 mM NaCl, 2 mM EDTA, 1% Triton X-100, 1× cOmplete protease inhibitor (Roche). Given fewer cells were used, only 20 µL of Dynabeads were used for the isolation of antibody-bound chromatin. The beads were washed in a series of three buffers the TE, twice each: wash buffer 1 [20 mM Tris pH 8.0, 150 mM NaCl, 2 mM EDTA, 1% Triton X-100, 0.15% SDS], wash buffer 2 [20 mM Tris pH 8.0, 500 mM NaCl, 2 mM EDTA, 1% Triton X-100, 0.1% SDS], wash buffer 3 [20 mM Tris pH 8.0, 250 mM LiCl, 2 mM EDTA, 0.5% Igepal CA-630, 0.5% sodium deoxycholate], and TE buffer [10 mM Tris pH 7.5, 1 mM EDTA]. After pulldown, the reversal of cross-links, DNA purification was as per above.

CTCF ChIP-seq libraries were prepared from ~15 million NSCs using the ChIP-IT High Sensitivity kit (Active Motif), according to the manufacturer's instruction, and 10 µg of CTCF antibody (Cell Signalling Technologies, D31H2).

Libraries were generated with an Illumina TruSeq DNA Sample Preparation Kit. 200- to 400-bp fragments were size-selected with AMPure XP magnetic beads. Libraries were quantified with a D100 tape in a 4200 Tapestation (Agilent). Libraries were pooled and sequenced on the Illumina NextSeq platform, with 75-bp single-end reads.

## ChIP-seq analysis

Adapter trimming was performed with TrimGalore! v0.4.4 with Cutadapt v.1.15[78], library QC was assessed with FastQC v0.11.8 before mapping with Bowtie2 v2.3.4.1[79] and Samtools v1.7[80] with default parameters to the GRCm38.p6 version of the reference mouse genome. BAM files were imported into SeqMonk v1.45.1 extending them by 150 bp to better represent size-selected fragments and peaks were called with the MACS style caller within the SeqMonk package (settings for 300 bp fragments, $P < 1 \times 10^{-5}$) by merging all three Smchd1GFP biological replicates and both WCE biological replicates into replicate sets. ChIP-seq tracks for the GFP ChIP were produced with SeqMonk by defining probes with a running window (width, 1000 bp; step 250 bp), doing a read-count quantitation then normalizing by library size before doing a match distribution normalization within each replicate set and smoothing over 5 adjacent probes. For the Smchd1-GFP ChIP scatter plot, the same data was quantified only over the peaks (±5 kb) published in Jansz et al.[14], normalized by total library size log2-transformed counts with datasets from merged biological replicates (three for each genotype). The H3K27me3 plot was produced the same way, using the lists of peaks from both genotypes produced as described above ±2.5 kb, then merging any probes closer than 1 kb.

CTCF ChIP-seq data was processed and mapped as described above, although using Cutadapt v2.9. SeqMonk v1.48.1 was used to read in BAM files (extending by 180 bp for all libraries as determined by cross-correlation plots using CSAW[81,82] v1.30.1 in R v4.2.0) and call peaks over each replicate set of libraries. Further processing was performed with CSAW, computing reads extended by 180 bp over a running window (width 20 bp, step 10 bp), then filtering over the previously determined peak regions so as to limit the subsequent more granular analysis to confirmed binding sites for at least one replicate set (peaks for each replicate set were merged into one list, overlapping regions were merged into single domains). Data was normalized with factors calculated with edgeR[83] v3.38.4 using the trimmed mean of $M$ values method in data binned at 10kbp. Differential binding in 20bp-windows was assessed with glmQLFTest (edgeR), comparing either $Smchd1^{MD43-GFP/MD43-GFP}$ or $Smchd1^{del/del}$ to $Smchd1^{GFP/GFP}$ data (all from the C57Bl/6 genetic background). Differential 20bp-windows were then merged if within 100 bp to a maximum length of 5kbp to be consistent

with the narrow nature of CTCF binding sites, merging p values with combineTests (CSAW), with an FDR (Benjamini-Hochberg) cutoff of 0.1. These results were then imported into SeqMonk to create MA plots highlighting peaks overlapping with said windows in each comparison.

### Reduced representation bisulfite sequencing (RRBS)

Genomic DNA was extracted using a Qiagen DNeasy kit. RRBS was performed with 100 ng of genomic DNA from each sample of female primary *Smchd1*[MommeD43/MommeD43] and *Smchd1*[+/+] NSCs (three independent cell lines of each genotype) with the Ovation RRBS Methyl-seq kit (NuGen) following the manufacturer's instructions. The bisulfite conversion was performed with the EpiTect Fast Bisulfite Conversion kit (Qiagen) following the instructions.

The libraries produced were cleaned and size-selected with AMPure XP magnetic beads. Libraries were quantified with a D100 tape in a 4200 Tapestation (Agilent). Libraries were pooled and sequenced on the Illumina NextSeq platform with 75-bp paired-end reads[84].

### RRBS analysis

The fastq files were processed with TrimGalore! v0.4.4 with Cutadapt v1.15[78] to remove Illumina adapter sequences. Custom adapter sequences from the Ovation RRBS kit were removed with the trimRRBSdiversityAdaptCustomers.py script provided on the manufacturer's website. Trimmed libraries were processed with Bismark v0.19.0[85] with Bowtie2 v2.3.4[79] and Samtools v1.7[80]. They were aligned to the GRCm38.p6 version of the mouse genome.

The bismark coverage files produced were then loaded into R v3.6.1 and processed using the methylKit v1.10 package[86] (bismark-Coverage pipeline for CpG context), filtered by coverage (minimum 10 reads per base, maximum 99.9% percentile). The reported differences in methylated bases were obtained with the getMethylDiff function for a difference of at least 20% in methylation status with q-value < 0.05 for significance.

### In situ Hi-C

Three independent female NSC cell lines of each *Smchd1*[MommeD43-GFP/MommeD43-GFP] and *Smchd1*[GFP/GFP] genotype were used to generate in situ Hi-C libraries exactly as previously described[14] based on Rao et al.[87], using MboI as a restriction enzyme. Libraries were sequenced on the Illumina NextSeq platform, with 75-bp paired-end reads.

Primary-data processing was performed with HiCUP v0.7.2[88], mapping reads to the mm10 mouse genome assembly. DIs were identified with diffHiC v1.16.0[89], which uses edgeR statistics[83] (edgeR v3.26.8 and R v3.6.1). In brief, reads mapped and filtered with HiCUP were counted into 100-kb and 1-Mb bin pairs. The noise was removed by filtering out low-abundance reads on the basis of a negative binomial distribution and with interchromosomal counts to determine nonspecific ligation events. Libraries were then normalized with LOESS normalization (the counts from the matrices' diagonals were normalized separately to the rest), and trended biases were removed by fitting libraries to a generalized linear model. EdgeR was then used to test for differential interactions between genotypes, at either 100-kb or 1-Mb resolution, with a quasi-likelihood F test, and then adjustment was performed for multiple testing with FDR. To make a track of significant differential interactions for chromosome X, both sets of anchors were merged into a single 2D track.

### Generation of Hi-C contact matrices

To construct the Hi-C interaction matrices of the *Smchd1*[MommeD43-GFP/MommeD43-GFP] and *Smchd1*[GFP/GFP] genomes, we used TADbit v0.4.39[90] with the original iterative mapping strategy ICE (mm10 reference genome). The minimal size for mapping was set to 25 bp, and the iterative mapping procedure increased in steps of 5 bp until a maximal read length of 75 bp was reached. The reads were

filtered with the apply_filter function from TADbit with the following parameters: maximum molecule length adjusted individually for each library to the longest insert in the 99.9% percentile, minimum distance to restriction site to be defined as a random break equal to 1.5 times the maximum molecule length, minimum fragment size of 150 bp, too close to restriction site if within 4 bp of it and default settings for the other parameters. Once filtered, the read pairs were binned at 100-kb or 1-Mb resolution, and columns containing few interaction counts were removed according to the two-step strategy described in Serra et al.[90]. The remaining bins were further normalized with ICE as implemented in TADbit. After checking the correlation between the 3 biological replicates of each genotype (Spearman correlation >0.98), we merged the unnormalized reads into a single dataset for each of them. The new datasets were then normalized the same way.

### TAD detection on Hi-C contact maps

The merged interaction maps were used for domain detection at a 100-kb resolution. The TADbit program uses a breakpoint-detection algorithm that returns the optimal segmentation of the chromosome with a BIC-penalized likelihood[91]. TADbit was used to make the TAD alignment diagrams for chromosome 3, 6, 11, and X at 100 kb resolution.

### Compartment detection on Hi-C contact maps

The merged interaction maps were used for compartment detection at 1-Mb resolution. The TADbit program was used to search for compartments in each chromosome by computing a correlation matrix on the normalized matrix and by using the first eigenvector to identify compartments. TADbit was used to make the Eigenvector-difference plot for chromosome X.

### Capture-C

Immediately after dissection in cold PBS, tailbud tissue was mechanically dissociated, then fixed with 1% formaldehyde (Sigma Aldrich) for exactly 10 min at room temperature. Formaldehyde was quenched with glycine to a final concentration of 0.2 M and the sample was incubated for 5 min at room temperature. The sample was then centrifuged at 3000 × g for 5 min, then resuspended in Hi-C lysis buffer (as per in situ Hi-C method) with cOmplete protease inhibitor (Roche) for 20 min on ice to extract intact crosslinked nuclei. After centrifugation at 5000 ×g for 5 min at 4 °C, the pellet was snap-frozen and stored at −80 °C. Upon thawing, 2–3 tailbud samples per genotype were pooled and in situ Hi-C was carried out as described above. Hi-C libraries were prepared using the NEBNext Ultra II DNA Library Prep Kit for Illumina (New England Biolabs) kit according to the manufacturer's instructions and amplified with 9 PCR cycles. Samples were then quantified with a high-sensitivity D1000 Tapestation (Agilent Technologies) and high-sensitivity dsDNA Qubit and then pooled at equimolar ratios. Capture-C was carried out on the pooled libraries using the Agilent SureSelect Target Enrichment Kit (Roche) according to the manufacturer's instructions. Baits for SMCHD1 targets of interest were designed using the online Agilent Technologies SureDesign Custom Design Tool. The captured pool was then sequenced on the Illumina NextSeq platform with 75 bp paired-end reads.

### Capture-C analysis

Reads were mapped using HiCUP v0.8.3 to the mm10 assembly of the mouse reference genome, providing a DpnII-digested version of the same. BAM files were then imported into CHiCANE[56] v0.1.8 in R v4.2.0. DpnII restriction fragments of the genome were intersected with coordinates of regions covered by capture probes, limiting to unique fragments of at least 20 bp. Replicates were merged (method='sum') and significant interactions were determined with an FDR cutoff of 0.05, default parameters (negative binomial distribution model with overdispersion term by maximum likelihood over all pairs of

fragments, the fitted model is used to estimate a *P* value for each pair, corrected for multiple testing with Benjamini–Hochberg method).

Heatmaps were generated using the cloud-base implementation of Juicebox[92] v2.2.6 after merging BAM files from replicates using samtools merge v1.17, and converting them to.txt format with a script provided in HiCUP then to.hic format by the *pre* function from Juicer_tools[93] v2.20.00.

## RNA-sequencing

For RNA-sequencing in primary NSCs of the following genotypes: *Smchd1^MommeD43/MommeD43* vs *Smchd1^+/+* (FVB/NJ background) and *Smchd1^MommeD43-GFP/MommeD43-GFP* vs *Smchd1^GFP/GFP* (C57BL/6) cells from the indicated number of independent lines were harvested at an early passage (£10) and RNA was extracted and treated with DNAse I with the Quick RNA kit (Zymo) as per the instructions. For PSM RNA-sequencing libraries, tissue was dissected from E8.5 embryos and E10.5 embryos respectively, as described in the relevant methods sections, snap-frozen, then treated the same way as the other samples. 100 ng of total RNA was used to make libraries with the TruSeq RNA Library Prep kit v2 or the TruSeq stranded mRNA Library Prep kit following the manufacturers' instructions. 200- to 400-bp products were size-selected and cleaned up with AMPure XP magnetic beads. Final cDNA libraries were quantified with a D1000 or D5000 tape in the 4200 Tapestation (Agilent) and pooled for sequencing on the Illumina NextSeq platform, with 75-bp single-end reads.

## RNA-sequencing analysis

Reads from the experiments mentioned above as well as previously published data from Chen et al. PNAS 2015 (*Smchd1^MommeD1/MommeD1* vs *Smchd1^+/+* RNA-seq in NSCs of FVB/NJ background; data available from GEO under accession number GSE65747) were trimmed for adapter sequences with TrimGalore! v0.4.4 with Cutadapt v2.9[78] then mapped with hisat2 v2.0.5[94,95] to the GRCm38.p6 reference assembly of the mouse genome.

For the E8.5 PSM libraries and the E10.5 FNP libraries, the analysis was performed using SeqMonk v1.47.2 by importing the BAM files and using its RNA quantitation pipeline to create log2-transformed counts normalized by library size and corrected for transcript length to obtain log2 RPKM counts. Heatmaps of the log2 fold-change were made by subtracting the log2 RPKM counts from the control to the test dataset. Differential gene expression analysis on FNP samples was performed with SeqMonk's inbuilt EdgeR-based statistical test[83] with an FDR cutoff of 0.05 after excluding sex chromosomes.

Analysis of NSC RNAseq data from *Smchd1^MommeD43/MommeD43* vs *Smchd1^+/+*, *Smchd1^MommeD43-GFP/MommeD43-GFP* vs *Smchd1^GFP/GFP* and *Smchd1^MommeD1/MommeD1* vs *Smchd1^+/+* was performed using limma v3.40.6[96] with edgeR v3.26.8[83] in R v3.6.1. Since these data were obtained from multiple experiments, we corrected for batch effects using technical replicates across batches as well as sex (all *Smchd1^MommeD1/MommeD1* cells are male since *MommeD1* is an embryonic lethal mutation in homozygous females) and strain (FVB/NJ or C57BL/6). Linear models were fitted to the log2 CPM count matrices using voom with quality weights[97]. DEGs were determined using empirical Bayes' moderated *t* tests[98] cutoff of 0.05.

## Gene ontology analysis

GO term analysis of the differentially expressed genes found in *Smchd1^MommeD43/MommeD43* FNP RNA-seq was performed by uploading a list of those 56 genes to the geneontology.org GO Enrichment Analysis platform, looking for affected biological processes in *Mus musculus*[99].

## Reporting summary

Further information on research design is available in the Nature Portfolio Reporting Summary linked to this article.

## Data availability

The genomics data discussed in this publication have been deposited in NCBI's Gene Expression Omnibus[100] and are accessible through GEO SuperSeries accession number GSE174113. All other data used to construct figures are provided as Source Data. Source data are provided with this paper.

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

## Acknowledgements

We thank Dr. Ruth Arkell for the valuable discussion of the FNP RNA-seq and the WEHI MAGEC facility for recreating the *MommeD43* mutation on the *Smchd1-GFP* background. We thank mouse technicians at WEHI (Jessica Martin) and QIMR (Joanne Sutton) for their work on this project. We thank SAXS beamline staff at the Australian Synchrotron for assistance with data collection. This work was supported by grants and fellowships from the Australian National Health and Medical Research Council (GNT1098290 to M.E.B., J.M.M., and M.E.R., fellowship GNT1194345 to M.E.B., GNT1172929 to J.M.M., GNT1104924 to M.E.R.). N.J. and A.G. were supported by Australian Research Training Program scholarships. MEB was supported by the Bellberry-Viertel fellowship. J.C.G. was funded by a fellowship from the FSHD Society (Sylvia & Leonard Marx Foundation Fellowship FSHS-82017-02). This work was also supported by grants from the Prinses Beatrix Spierfonds (W.OP14-01, W.OR17-04), the US National Institutes of Health (National Institute of Arthritis and Musculoskeletal and Skin Diseases 2R01AR066248), and Spieren voor Spieren. The generation of the *MommeD43* mice on the *Smchd1-GFP* background used in this study was supported by Phenomics Australia (PA) and the Australian Government through the National Collaborative Research Infrastructure Strategy (NCRIS) program. Additional support was provided by the Victorian State Government Operational Infrastructure Support, Australian National Health and Medical Research Council IRIISS grant (9000719). The Australian

Regenerative Medicine Institute is supported by grants from the State Government of Victoria and the Australian Government. S.X. is supported by NMRC (NMRC/OFYIRG/062/2017) and NUS PYP. B.R. is an investigator of the National Research Foundation (NRF, Singapore), Branco Weiss Foundation (Switzerland), and an EMBO Young Investigator, and is supported by an inaugural Use-Inspired Basic Research Fund from the Agency for Science & Technology and Research (A*STAR) in Singapore. B.H., Y.K., S.M., and J.G. are members of the European Reference Network for Rare Neuromuscular Diseases [ERN EURO-NMD].

## Author contributions

A.T.d.F., B.d.H., N.B., N.J., K.C., T.B., H.V., A.D.G., L.D., S.X., T.T.N.L., I.W., M.I., K.B., H.O., Y.D.K., D.vd.H., L.F.B., Q.A.G., F.P., E.M., and A.K. performed experiments and analyzed data. T.M.J. and T.M. provided expert advice. M.E.R., B.R., S.X., S.vd.M., E.M., J.M.M., J.d.G., and M.E.B. provided expert advice, supervised the work, and provided funding. A.T.d.F. and J.G. drafted the manuscript. All authors edited the manuscript.

## Competing interests

The authors declare no competing interests.
