## [Peer Review File · Nature Communications]

SMCHD1 has separable roles in chromatin architecture and gene silencing that could be targeted in diseaseREVIEWER COMMENTS

Reviewer #1 (Remarks to the Author):

The authors described a mutant form of SMCHD1 (MommeD43) with enhanced transgene silencing activity. The MommeD43 mutation did not affect stability of the SMCHD1 protein, nor its binding to target loci by ChIP-seq analysis. The mutation was in exon 15 of the *Smchd1* gene, within the extended ATPase domain of SMCHD1. Structure and function analysis revealed that the mutation neither altered the conformation nor activity of the ATPase domain. For phenotypic analysis, they demonstrated that the homozygous mutant mice exhibited an anterior homeotic transformation with additional effects on rib formation possibly due to *Hoxa6* and *Hoxd10* gene silencing. They further validated the increased transgene silencing activity by crossing the mutant mice with hemizygous D4Z4-2.5 mice, which carrying a human transgene consisting of 2.5 D4Z4 repeat units cloned from genomic DNA of an FSHD1 patients. Finally, they tried to understand the mechanism behind the apparent gain-of-function of the mutant SMCHD1 at 3D genome level by in situ Hi-C analysis. The phenotype is interesting, but the mechanism is incomplete.

Specific comments

- 1) The authors claimed that MommeD43 has a gain of function effect on Hox gene silencing and skeletal development. There is no data showing whether SMCHD1 binds the promoter region of the *Hoxa6* and *Hoxd10* genes.
- 2) They identified 53 downregulated genes in frontonasal prominence (FNP) of the homozygous MommeD43 mutant embryos. It would be helpful to indicate those genes are direct or indirect targets of the SMCHD1.
- 3) In FSHD mice model, they found that *Dux4* expression in skeletal muscle were comparable between D4Z4-2.5/*Smchd1*^{+/+} and D4Z4-2.5/*Smchd1*-MommeD43/+ mice. However, it was significantly decreased in primary myoblasts and differentiated myotubes from D4Z4-2.5/*Smchd1*-MommeD43/+ mice compared to D4Z4-2.5/*Smchd1*^{+/+} controls. How do they think about such discrepancy? What's like the expression pattern of *Dux4* gene in proliferating myoblasts, differentiated myotubes and mature myofibers?
- 4) The mechanisms were very complicated. Because the phenotypes of the MommeD43 mutant mice involved multiple cell lineage, including progenitors of the vertebral column, frontonasal prominence (FNP), skeletal muscle. I wonder why they performed Hi-C analysis in neural stem cells rather than using progenitors of the vertebral column, frontonasal prominence (FNP) or skeletal muscle.
- 5) Recently, increasing lines of evidence showed that there is distinct 3D genome architecture in different cell lineages. Thus, it is very hard using Hi-C data in neural stem cells to explain gene expression observed in progenitors of the vertebral column or skeletal muscle.
- 6) For Hi-C data analysis, they specifically focused on chromatin interactions between *Hoxb* and keratin locus, between *Hoxb* and olfactory receptor gene. Are those genes associated with the observed phenotype in the MommeD43 mutant mice?

Minor points

There is no standard error bars in Figure 4?

Reviewer #2 (Remarks to the Author):

In the comprehensive study by Fierro et. al. they used two *Smchd1* mutant mice to characterize the function of the A667E mutant. They demonstrated the missense mutation does not affect protein levels, known binding sites or ATPase domain structure or activity, concluding that the impact must be on protein function.

They were able to demonstrate a developmental phenotype due to altered regulation of the Hox cluster, which is a known target for *Smchd1*. The mouse, however, did not show any overlapping morphological phenotypes with BAMS. Interesting, the cross of this mouse with the D4Z4-2.5

mouse was able to reduce DUX4 expression and the biomarker Wfdc3. Lastly, they performed Hi-C experiments and observed results similar to the Smchd1 knock out mouse.

The authors concluded that the A667E mutant is a neomorph with altered function, due to its gain of function in terms of transcription/insulation and hypomorph feature in chromatin architecture.

1. A major concern in this manuscript is that two different Smchd1 mice were used. The first is from an ENU screen (MommeD43) and another from CRISPR modification of GFP-Smchd1 mouse to have the same A667E mutation. The claim is that these mice can be used "interchangeably for the remainder of this study."

The identification of the A667E mutation in the MommeD43 based on referencing was identified in 2013 and the authors in this study did not attempt to further characterize the linkage peak any further using more modern techniques that includes whole genome sequencing (WGS). WGS at adequate depth will rule out any non-coding or other coding variants in the linkage peak or outside that may further contribute to functional results observed in this study.

Are there any other genes within the chromosome 17, 63-98 Mb linkage region that have relevant functions?

It is absolutely critical to be transparent in the discussion which of the functional results (transcription/insulation vs chromatin architecture) is from which mouse and the implications this may have on the interpretation of the results, if any. Furthermore, this is even more important due to the conclusions of this study regarding Smchd1 function.

2. It is still unclear how DUX4 down-regulation is achieved in the D4Z4-2.5/MommeD43 mouse and the authors need to be clearer in this section.

a. The authors state the following:

"In humans, silencing of the D4Z4 repeat is achieved by DNA methylation and repressive chromatin marks such as histone 3 lysine 9 trimethylation (H3K9me3) and histone 3 lysine 27 trimethylation (H3K27me3)"

The experiments showed no difference in methylation or chromatin marks. The authors then later introduce chromatin compaction score as the possible mechanism observed previously. It needs to be clearer, which of these three features are important to FSHD and the expression of DUX4 and its relevance to the human disease.

b. In Figure 4f, it is unclear which of the results are from fibroblast and which are from spleen or if the results were aggregated. If fibroblast results were included, then the authors need to present the impact on DUX4 expression on the RNA and protein level similar to what was shown for myoblast/myotubes and spleen. This was not present in Supplementary Figure 7.

c. It is unclear from the results section whether the chromatin marks that were determined were localized to the D4Z4 region similar to observing methylation or was this a global whole genome measure? The methods suggest it is targeted to D4Z4, but it would be good to be explicit and clear.

d. The authors need to revise their claims on the translational aspects of their results.

(i) They current results show the impact this mutation has in a mouse model that includes a human D4Z4 in the wrong genomic context as it does not lie on the sub-telomeric region.

(ii) Furthermore, they have used this mouse produced by ENU as a proxy for a SMCHD1 gain of function mutation. As discussed above the rigor of the genetic characterization given the technology is not of standard anymore.

(iii) The authors have not demonstrated this result in a control and FSHD patient cell lines. All results presented are from the mouse, a very imperfect model in isolation for human FSHD disease. This mouse does not have a muscle phenotype so would be the equivalent of claiming the

MommeD43 would be a good model for the BAMS phenotype based on its differential expression of relevant genes.

(iv) In the discussion the authors have written:

“The gain-of-function effects of the MommeD43 variant could provide a potentially elegant solution for treatment of the FSHD phenotype by revealing how to stimulate the wild-type copy of SMCHD1 that is retained in FSHD patients to silence DUX4.”

The authors did not use the wildtype but the mutant Smchd1. Also, why would you use the mutant with functions that is not fully characterized when you can replace with the wild type copy? Also, given SMCHD1 can bind to other places in the genome, the potential off target effects would need to be mentioned.

Reviewer #3 (Remarks to the Author):

In this manuscript, the authors characterized a chemically induced mouse mutant (MommeD43) of the chromatin-associated protein SMCHD1. This protein is mutated in two human diseases, facioscapulohumeral muscular dystrophy (FSHD) and Bosma arrhinia microphthalmia syndrome (BAMS). The mutation is found in the extended ATPase domain of the protein. The authors observed an anterior homeotic transformation phenotype of MommeD43 mice affecting the rib cage structures. Using different assays, they concluded that the mutant has mostly a gain of function phenotype with increased gene repression properties and increased shielding against PRC2-induced H3K27 methylation on the X chromosome. Some long-range effects seemed to be more consistent with a loss of function. The study has been done very logically and meticulously, and the data have been represented clearly. However, most of the results were negative leaving some questions as to the relevance of the findings for the human diseases and also as to the mechanism how this particular SMCHD1 mutant functions in chromatin.

Specific comments:

- 1) The authors did not specify if the human equivalent mutation of A667E has ever been found in BAMS or in FSHD.
- 2) Several of the experiments were performed with neural stem cells (NSCs), a cell type not directly relevant for the human diseases.
- 3) Figure 1:
What is the control in the SMCHD1 ChIP-seq experiments? What is WCE?
- 4) There is considerable information in the literature regarding a crosstalk between SMCHD1 and the CTCF insulator protein. Have the authors determined if CTCF binding is perturbed in the MommeD43 mutant?

Thank you to the reviewers for their helpful suggestions to improve the manuscript. We respond to each in turn, with our replies in blue text. We believe the additional experiments and explanations have improved the paper.

In summary we have included:

- Replication of the immunofluorescence experiment measuring H3K27me3 and SMCHD1 levels in the inactive X of female NSCs in the CRISPR allele (Figure 6.e-g).
- 3 new genomic experiments that further strengthen our work and address the reviewer questions:
 1. Capture-C on dissected embryonic presomitic mesoderm tissue, which required dissection of over 600 embryos to obtain the embryos of the required somite number and genotype for the somite-matched samples required. This work addresses the potential role of SMCHD1 in short-range interactions at *Hox* gene clusters, in tissues relevant to the phenotypes we observe and the role of the *Hox* clusters in development;
 2. ChIP-seq for CTCF in *Smchd1* gain of function mutant (MommeD43) and *Smchd1* null neural stem cell lines to identify changes in CTCF binding caused by the MommeD43 mutation, to enhance our understanding of how the gain of function mutant changes function at a mechanistic level;
 3. Nanopore sequencing to completely sequence through the linked interval for the original MommeD43 mutant line, strengthening the evidence for the *Smchd1* mutation being the causative mutation of the effects we observe.
- We have incorporated extensive edits related to all of the reviewer comments and included a new 'Limitations of our study' section in the discussion, and a supplementary table that summarises all the data from different assays to make our results clear for the reader.

REVIEWER COMMENTS

Reviewer #1 (Remarks to the Author):

The authors described a mutant form of SMCHD1 (MommeD43) with enhanced transgene silencing activity. The MommeD43 mutation did not affect stability of the SMCHD1 protein, nor its binding to target loci by ChIP-seq analysis. The mutation was in exon 15 of the *Smchd1* gene, within the extended ATPase domain of SMCHD1. Structure and function analysis revealed that the mutation neither altered the conformation nor activity of the ATPase domain. For phenotypic analysis, they demonstrated that the homozygous mutant mice exhibited an anterior homeotic transformation with additional effects on rib formation possibly due to *Hoxa6* and *Hoxd10* gene silencing. They further validated the increased transgene silencing activity by crossing the mutant mice with hemizygous D4Z4-2.5 mice, which carrying a human transgene consisting of 2.5 D4Z4 repeat units cloned from genomic DNA of an FSHD1 patients. Finally, they tried to understand the mechanism behind the apparent gain-of-function of the mutant SMCHD1 at 3D genome level by in situ Hi-C analysis. The phenotype is interesting, but the mechanism is incomplete.

Specific comments

1) The authors claimed that MommeD43 has a gain of function effect on *Hox* gene silencing and skeletal development. There is no data showing whether SMCHD1 binds the promoter region of the *Hoxa6* and *Hoxd10* genes.

Our previous SMCHD1 ChIP-seq analyses in NSCs (Chen et al., PNAS 2015; Jansz et al NSMB 2018) have shown broad SMCHD1 enrichment across all *Hox* clusters, including over *Hoxa6* and *Hoxd10*. In this paper, we have also performed *Smchd1* ChIP-seq in wild-type and MommeD43 NSCs and find similar enrichment. We have now specified that SMCHD1 is broadly enriched across all 4 *Hox* clusters including across the 2 genes where we observe gain of silencing (see page 6). These data are now provided in Supplementary Figure 4.

2) They identified 53 downregulated genes in frontonasal prominence (FNP) of the homozygous MommeD43 mutant embryos. It would be helpful to indicate those genes are direct or indirect targets of the SMCHD1.

Thank you for this suggestion. SMCHD1 ChIP-seq required 40M cells, so we can't do this in the FNP cell population. Instead, we have examined SMCHD1 ChIP-seq peaks within 5kb of these genes in our NSC

dataset. We observe that 20 of the 53 differentially expressed genes in MommeD43 FNP have SMCHD1 binding sites nearby in NSCs. We have added this analysis to the text and noted that from what we can conclude from using different cell populations, these genes appear to be more sensitive to MommeD43 than loss of SMCHD1 (see page 8).

3) In FSHD mice model, they found that Dux4 expression in skeletal muscle were comparable between D4Z4-2.5/Smchd1^{+/+} and D4Z4-2.5/Smchd1-MommeD43^{+/+} mice. However, it was significantly decreased in primary myoblasts and differentiated myotubes from D4Z4-2.5/Smchd1-MommeD43^{+/+} mice compared to D4Z4-2.5/Smchd1^{+/+} controls. How do they think about such discrepancy? What's like the expression pattern of Dux4 gene in proliferating myoblasts, differentiated myotubes and mature myofibers?

As previously described for D4Z4-2.5 mice (de Greef et al, Hum Mol Genet 2018), DUX4 expression levels are very low and variable in skeletal muscles (Cq values are between 34 and 38, even when 3 microgram of RNA is used as input for cDNA synthesis). It is therefore difficult to reliably establish differences in expression level, particularly a decrease in expression. For this reason, we believe that we may have been unable to find significant changes in DUX4 expression levels of skeletal muscles between D4Z4-2.5/Smchd1^{+/+} and D4Z4-2.5/Smchd1^{MommeD43/+} mice. As DUX4 expression levels are somewhat higher, and less variable, in cultured muscle cells of the D4Z4-2.5 mouse models (Cq values are between 31 and 33 in both proliferating myoblasts and differentiated myotubes), we have been able to detect significant changes in these cells. To clarify this, we have added these Cq values in the main text, section "MommeD43's improved repeat-silencing capability offers therapeutic potential for FSHD" (page 10/11), and we have rephrased this paragraph.

With regards to the expression of DUX4 in myoblasts, myotubes and myofibers, this has been reported in Krom et al., PLoS Genetics 2013. In this work, DUX4 positive nuclei were negative for Myog, suggesting DUX4 inhibits differentiation, as has been reported previously for zebrafish.

4) The mechanisms were very complicated. Because the phenotypes of the MommeD43 mutant mice involved multiple cell lineage, including progenitors of the vertebral column, frontonasal prominence (FNP), skeletal muscle. I wonder why they performed Hi-C analysis in neural stem cells rather than using progenitors of the vertebral column, frontonasal prominence (FNP) or skeletal muscle.

We chose NSCs originally to compare to our extensive published datasets on SMCHD1 null NSCs, rather than create all datasets in MommeD43 and recreate them for the null allele for comparison. We appreciate the reviewer seeking further clarification, and have added to our explanation in the text accordingly.

Given the vertebral column phenotype we observe in the MommeD43 mutants, and the change in *Hox* gene expression that is consistent with this effect at the ~8 somite stage in tailbuds containing the presomitic mesoderm, we now provide additional experimentation in this tissue. We performed Capture-C as we theorized that the comparatively low resolution of the HiC data in NSCs may preclude capturing short range interaction changes within each *Hox* cluster that may be more mechanistically revealing. Our new Capture-C in MommeD43 embryos – both on the C57BL/6J background (CRISPR allele) and the FVB background (ENU allele) – does not reveal any substantial changes in structure of each *Hox* gene cluster. Each shows the expected domains analogous to the C-domain and T-domain reported by the Duboule lab previously. It is important that while these data are high resolution compared with HiC, the quantification of differences is limited due to the small amount of DNA in the original samples. These new data are presented in Fig. 5j and Supp. Fig. 11. Importantly, these new genomic data in a tissue relevant to our phenotype have not altered our conclusions, but instead are consistent with SMCHD1 normally regulating long range interactions over megabase intervals, such as between *HoxB* and the olfactory receptor cluster illustrated by DNA FISH (see Fig. 5g-i).

We appreciate that it would appear simpler to perform all genomic analyses in the relevant tissue. The difficulties in collecting embryos of the right sex, genotype and somite-stage, as well as the very low number of cells preclude using presomitic mesoderm tissue or any freshly dissected tissue for most experiments, and we have now noted this in the text in the results (page 14) and in a new limitations section in the Discussion (page 21).

To further examine potential mechanistic explanations for how the MommeD43 mutation alters SMCHD1 function, given that we did not obtain any additional clues based on the Capture-C data, we went back to NSCs and performed CTCF ChIP-seq. Past work by us and others has shown a potentially competitive relationship between SMCHD1 and CTCF (Chen et al., PNAS 2015; Jansz et al., NSMB 2018; Wang et al., Cell 2018; Gdula et al., Nature Comms 2019). Here we have found relatively limited effects on CTCF binding in both null and MommeD43 samples (C57BL/6J background). However, for the peaks that are significantly differentially enriched (determined by CSAW, Supplementary Table 10), binding is enhanced in the null samples and depleted in the MommeD43 samples (Fig. 6c, Supp. Fig. 12). These data are consistent with the gain-of-function effects that we observe on gene expression on autosomes and insulation on the inactive X, rather than the hypomorphic effects on chromatin architecture observed by *in situ* HiC and DNA FISH. Therefore, we have retained our conclusion that the MommeD43 mutation impacts functions of SMCHD1 that do not relate to chromatin architecture.

5) Recently, increasing lines of evidence showed that there is distinct 3D genome architecture in different cell lineages. Thus, it is very hard using Hi-C data in neural stem cells to explain gene expression observed in progenitors of the vertebral column or skeletal muscle.

We agree. We originally used NSCs so that we could compare to the *Smchd1* null data that we already published in this cell type. As described above, we have now performed Capture-C in tailbud tissue, the progenitors of the vertebral column; however, we do not observe striking changes in the MommeD43 samples. We also performed Capture-C in MommeD1 samples, unfortunately we did not obtain good libraries for these samples, likely due to very low input, so were unable to compare between MommeD43 and the null samples.

The lack of an effect in Capture-C is perhaps not surprising, as Capture-C shows a focused, high-resolution chromatin architecture snapshot of short-range interactions within the studied clusters, whereas *in situ* HiC is genome-wide, lower resolution and suited to study of longer-range interactions. We therefore conclude that, taking into account the low input samples, SMCHD1 likely has a more prominent role in long-range rather than short-range interactions, consistent with prior work from us and others (Jansz et al., NSMB, 2018; Wang et al., Cell 2018). This is addressed on page 14/15 in the text and in the new limitations section in the Discussion (page 21), so it is clear for the reader.

6) For Hi-C data analysis, they specifically focused on chromatin interactions between *Hoxb* and keratin locus, between *Hoxb* and olfactory receptor gene. Are those genes associated with the observed phenotype in the MommeD43 mutant mice?

We do not specifically observe a phenotype related to the olfactory receptors or the keratin cluster of genes. We focused on the interaction of these regions with *Hoxb* for validation by DNA FISH as we needed to choose one interaction, and that was one of the most significantly different in our previously published null versus wild-type analysis (Jansz et al., NSMB, 2018). However, *in situ* HiC was genome-wide, and our analyses are shown as such. Our new Capture-C in tailbud focuses on the 4 *Hox* clusters, which do relate to the phenotype. To ensure this is clear for the reader, we have noted that presomitic mesoderm is relevant to phenotype on page 14 in the new Capture-C section.

Minor points

There is no standard error bars in Figure 4?

We chose not to show standard error bars in Figure 4, since each dot represents the data of a single mouse (as also mentioned in the legend), and as such immediately shows the variation between mice. By adding standard errors, we feel that the figures would be less clear.

Reviewer #2 (Remarks to the Author):

In the comprehensive study by Fierro et. al. they used two *Smchd1* mutant mice to characterize the function of the A667E mutant. They demonstrated the missense mutation does not affect protein levels, known binding sites or ATPase domain structure or activity, concluding that the impact must be on protein function.

They were able to demonstrate a developmental phenotype due to altered regulation of the *Hox* cluster,

which is a known target for *Smchd1*. The mouse, however, did not show any overlapping morphological phenotypes with BAMS. Interestingly, the cross of this mouse with the D4Z4-2.5 mouse was able to reduce DUX4 expression and the biomarker *Wfdc3*. Lastly, they performed Hi-C experiments and observed results similar to the *Smchd1* knock out mouse.

The authors concluded that the A667E mutant is a neomorph with altered function, due to its gain of function in terms of transcription/insulation and hypomorph feature in chromatin architecture.

1. A major concern in this manuscript is that two different *Smchd1* mice were used. The first is from an ENU screen (MommeD43) and another from CRISPR modification of GFP-*Smchd1* mouse to have the same A667E mutation. The claim is that these mice can be used "interchangeably for the remainder of this study."

The identification of the A667E mutation in the MommeD43 based on referencing was identified in 2013 and the authors in this study did not attempt to further characterize the linkage peak any further using more modern techniques that includes whole genome sequencing (WGS). WGS at adequate depth will rule out any non-coding or other coding variants in the linkage peak or outside that may further contribute to functional results observed in this study.

It is absolutely critical to be transparent in the discussion which of the functional results (transcription/insulation vs chromatin architecture) is from which mouse and the implications this may have on the interpretation of the results, if any. Furthermore, this is even more important due to the conclusions of this study regarding *Smchd1* function.

We thank the reviewer for raising these important points. The original mapping and exome sequencing did occur in 2013 as the reviewer rightly observes, although this is the first publication on this mouse model. We used SNP chip followed by exome sequencing and this has now been made clearer in the text (page 3). As rightly noted, we cannot rule out non-coding variants with this approach. Based on the reviewer's suggestion we have now used Nanopore long-read whole genome sequencing. With 20-30X coverage using this approach, we have found that the *Smchd1* exon 15 mutation is the only coding variant in the linked interval. We identified 112 other single nucleotide variants, 9 of which were in putative regulatory elements. None of these were predicted to be damaging based on currently available software. These new data are included in Supplementary Table 1.

We have observed consistent effects with the original ENU mutant line and the recreated allele whenever it has been feasible to perform the experiments in both scenarios, taking into account the large strain-specific differences (FVB vs C57). This suggests that this mutation is sufficient for the effects we observe. These effects are those that we focus on and relate to our conclusions including gene expression in NSCs (RNA-seq in both strains), chromatin conformation (DNA FISH in both strains, Capture-C in both strains) and insulation (by H3K27me3 IF in both strains).

We appreciate that it is important to delineate the results from each allele and thank the reviewer for prompting inclusion of further explanation. To make it easier to interpret we have now specified in the text which allele contributed which result, and importantly where both alleles have been used. Of most importance, in all cases where it was possible to use both alleles, they show similar results. In summary:

1. Gene expression in NSCs in both backgrounds.
2. DNA FISH in both backgrounds.
3. H3K27me3 in both backgrounds.
4. Capture C in PSMs in both backgrounds
5. FNP RNAseq and PSM RNAseq only ENU induced on FVB background
6. Skeletal phenotype, only ENU induced on FVB background
7. D4Z4-2.5 mice, only ENU induced on FVB background
8. HiC, DNA methylation and ChIPseq only C57 CRISPR mutant

It is not feasible to perform all genomic analyses using both strains, due to cost. Moreover, we needed to perform the ChIP-seq studies for SMCHD1 in the C57 background as it has the SMCHD1-GFP which allows ChIP for SMCHD1, which is not possible with a SMCHD1 protein specific antibody. We therefore chose to

perform genomic analyses using these matched samples in NSCs and because NSCs provide a uniform cell type that is easily expanded while remaining diploid. This explanation is now expanded in the text on page 4 and page 14.

For the *in vivo* experiments, FVB provides larger litter sizes, fewer resorptions, and also more uniformly patterned embryos than C57BL/6, making the *in vivo* work much more feasible in this background. This has also been noted in the text (page 4). It is important to note that C57BL/6J skeletal patterning is much more variable than FVB/NJ.

We have now included a summary table of the experiments performed in each line (Supplementary Table 2), noted which allele was used throughout the text and a new 'Limitations of our study' section in the discussion to make it transparent for the reader.

Are there any other genes within the chromosome 17, 63-98 Mb linkage region that have relevant functions? Again, we appreciate the reviewer prompting further detailed analyses of this important point. We have now sequenced at 20-30X coverage the whole interval with Nanopore sequencing and the MommeD43 mutation is the only exonic mutation in this linked interval. As noted above, the variants observed in putative regulatory in the linked interval are not predicted to be detrimental and therefore our data support the role of the specific MommeD43 variant in causing the effects we present, especially given the positive data obtained in both the ENU mutant and the CRISPR mutant is entirely concordant.

2. It is still unclear how DUX4 down-regulation is achieved in the D4Z4-2.5/MommeD43 mouse and the authors need to be clearer in this section.

We agree that this is an important consideration and have now added a sentence in the main text, section "MommeD43's improved repeat-silencing capability offers therapeutic potential for FSHD" (page 11) to clarify this. Our previous studies have shown that the D4Z4 transgene in D4Z4-2.5 mice behaves like a FSHD allele in terms of DNA methylation, histone modifications and SMCHD1 binding (Krom et al., PLoS Genet 2013; de Greef et al., Hum Mol Genet 2018). Since the molecular mechanism by which SMCHD1 suppresses DUX4 is unknown, we focused on known chromatin features of the D4Z4 repeat in healthy individuals and FSHD patients to test the hypothesis that the MommeD43 mutation affects any of these features. We see an increase in H3K9me3 and a decrease in H3K4me2, suggesting that an alteration to the chromatin state may relate to the down-regulation of DUX4.

a. The authors state the following:

"In humans, silencing of the D4Z4 repeat is achieved by DNA methylation and repressive chromatin marks such as histone 3 lysine 9 trimethylation (H3K9me3) and histone 3 lysine 27 trimethylation (H3K27me3)"

The experiments showed no difference in methylation or chromatin marks. The authors then later introduce chromatin compaction score as the possible mechanism observed previously. It needs to be clearer, which of these three features are important to FSHD and the expression of DUX4 and its relevance to the human disease.

To better clarify the role of DNA methylation and chromatin marks in FSHD, we have added to the main text, section "MommeD43's improved repeat-silencing capability offers therapeutic potential for FSHD", explaining that H3K9me3 levels are reduced in all FSHD patients while H3K27me3 levels are specifically increased in FSHD2 patients. In addition, we have added some text to better explain the chromatin compaction score and its use in the FSHD field (page 12).

b. In Figure 4f, it is unclear which of the results are from fibroblast and which are from spleen or if the results were aggregated. If fibroblast results were included, then the authors need to present the impact on DUX4 expression on the RNA and protein level similar to what was shown for myoblast/myotubes and spleen. This was not present in Supplementary Figure 7.

We apologize if this was not clear in our original submission. Results from fibroblasts are depicted as black dots and results from the spleen as grey dots (see Figure 4 legend). A single dot represents data from a single mouse. We have now added DUX4, Wfdc3, and Smchd1 RNA expression data for these fibroblast samples in Supplementary Figure 8 (formerly Supplementary Figure 7). The currently available DUX4

antibodies do not work on fibroblast cultures and therefore we are unable to show DUX4 data at the protein level.

c. It is unclear from the results section whether the chromatin marks that were determined were localized to the D4Z4 region similar to observing methylation or was this a global whole genome measure? The methods suggest it is targeted to D4Z4, but it would be good to be explicit and clear.

Again, we thank the reviewer for their careful reading of our manuscript. We have now added to the main text, section “MommeD43’s improved repeat-silencing capability offers therapeutic potential for FSHD”, that the chromatin marks were determined at the D4Z4 repeat using a site-specific qPCR approach.

d. The authors need to revise their claims on the translational aspects of their results. (i) They current results show the impact this mutation has in a mouse model that includes a human D4Z4 in the wrong genomic context as it does not lie on the sub-telomeric region.

As with any animal model, the D4Z4-2.5 mice have their limitations as pointed out by this reviewer. However, in the absence of an endogenous D4Z4 repeat in the mouse (the equivalent Dux repeat is not subtelomerically located and it is unknown how its regulation compares to human D4Z4), this is currently the best animal model to use for our studies. Despite the fact that the D4Z4 transgene is not subtelomerically integrated in D4Z4-2.5 mice, we have demonstrated that its chromatin structure is comparable with that found in FSHD patients with reduced DNA methylation, a reduced chromatin compaction score and SMCHD1 binding. The absence of muscle pathology may be explained by the observation that activated satellite cells, once expressing DUX4, fail to participate in fusion, which is different from human muscle cells (Krom et al., PLoS Genet 2013). See also point (iv).

We have now included a limitations of our study paragraph in the discussion to mention the most relevant points.

(ii) Furthermore, they have used this mouse produced by ENU as a proxy for a SMCHD1 gain of function mutation. As discussed above the rigor of the genetic characterization given the technology is not of standard anymore.

As we have detailed above, it was not feasible to create all data with both alleles. Using Nanopore sequencing we have sequenced the whole of the linked interval at 20-30X coverage and do not find any other variants that are predicted to be detrimental. Taken together with us recreating the point mutation on a separate strain background and finding consistent positive results between the ENU-induced mutation and the recreated CRISPR mutation (by DNA FISH for long range interactions, IF for H3K27me3 and RNA-seq), we believe the evidence is compelling that the A667E mutation is a gain of function mutation in SMCHD1. Of course, in the future it will be revealing for additional mouse models to be studied where the underlying gain of biochemical activity is known e.g. gain of ATPase activity, however this is beyond the scope of the current work.

(iii) The authors have not demonstrated this result in a control and FSHD patient cell lines. All results presented are from the mouse, a very imperfect model in isolation for human FSHD disease. This mouse does not have a muscle phenotype so would be the equivalent of claiming the MommeD43 would be a good model for the BAMS phenotype based on its differential expression of relevant genes.

As acknowledged in the response to (i) above, all animal models are imperfect. However, for BAMS it is not yet clear which genes are relevant, whereas for FSHD this is known, and the gene in question has been introduced into these mice. While it is not a model of muscular dystrophy, it is the accepted model when considering the epigenetic silencing of D4Z4 in a mouse context (see Krom et al., PLoS Genet 2013; de Greef et al., Hum Mol Genet 2018.) This is mentioned in the new limitations section.

We have not yet been able to introduce the point mutation into control and FSHD patient-derived cells. This is a challenge when considering that we do not want to have any SMCHD1 deleted samples, which is the most common outcome even with homology directed repair. This will be undertaken in future studies.

(iv) In the discussion the authors have written:

“The gain-of-function effects of the MommeD43 variant could provide a potentially elegant solution for

treatment of the FSHD phenotype by revealing how to stimulate the wild-type copy of SMCHD1 that is retained in FSHD patients to silence DUX4.”

The authors did not use the wildtype but the mutant *Smchd1*. Also, why would you use the mutant with functions that is not fully characterized when you can replace with the wild type copy? Also, given SMCHD1 can bind to other places in the genome, the potential off target effects would need to be mentioned. We take the reviewer’s point and, based on the reviewer’s feedback, we have altered the wording of our discussion to be more specific about what we think can be learnt from these data about how SMCHD1 may be targeted to treat FSHD (see page 20). We believe it may be possible to learn how the MommeD43 mutation alters SMCHD1 and mimic this effect using a drug that targets wild-type SMCHD1. At this stage our evidence suggests that altering SMCHD1’s role in chromatin architecture may not achieve this effect, but functions related to insulation and CTCF binding may prove productive. The work presented here suggests it might be valid to mimic the effect of MommeD43, which was not previously known.

Reviewer #3 (Remarks to the Author):

In this manuscript, the authors characterized a chemically induced mouse mutant (MommeD43) of the chromatin-associated protein SMCHD1. This protein is mutated in two human diseases, facioscapulohumeral muscular dystrophy (FSHD) and Bosma arrhinia microphthalmia syndrome (BAMS). The mutation is found in the extended ATPase domain of the protein. The authors observed an anterior homeotic transformation phenotype of MommeD43 mice affecting the rib cage structures. Using different assays, they concluded that the mutant has mostly a gain of function phenotype with increased gene repression properties and increased shielding against PRC2-induced H3K27 methylation on the X chromosome. Some long-range effects seemed to be more consistent with a loss of function. The study has been done very logically and meticulously, and the data have been represented clearly. However, most of the results were negative leaving some questions as to the relevance of the findings for the human diseases and also as to the mechanism how this particular SMCHD1 mutant functions in chromatin.

Specific

comments:

1) The authors did not specify if the human equivalent mutation of A667E has ever been found in BAMS or in FSHD.

We thank the reviewer for this useful suggestion. We have now specified that the same mutation has never been reported in BAMS (see page 8). It is worth noting that even though this is not a patient mutation, it is still the first mutant studied with apparent gain of function effects with extensive genomic and developmental characterisation.

2) Several of the experiments were performed with neural stem cells (NSCs), a cell type not directly relevant for the human diseases.

This is true, however it is not always possible to perform genomic analyses, which require high inputs, in cell types relevant to disease. We therefore used NSCs because we have previously published many genomic datasets in *Smchd1* null NSCs that were useful for comparison when considering the mechanism of MommeD43. We have noted this now in the text for clarity (see page 14).

We have now added new Capture-C in tailbud, relevant to the *Hox* gene expression change and relevant vertebral column phenotype (see page 14/15, Fig. 5j). We have also measured chromatin state and expression at the D4Z4-2.5 transgene in muscle as well as other tissues, which is relevant to FSHD (see Fig. 4). Finally, we measured expression in FNPs relevant to BAMS (see Fig. 3).

So, while it wasn’t feasible to perform all studies in disease-relevant tissues, we have chosen each tissue carefully for the relevant experiments. We have added more explanation into the text for why we chose to use NSCs for most genomic analyses, which was to compare to our *Smchd1* null data and to make our analyses feasible with a bulk population of uniform cells (see page 14).

3) Figure 1:

What is the control in the SMCHD1 ChIP-seq experiments? What is WCE?

The control for these experiments is the whole cell extract (WCE), which is the sample that has been through all stages of ChIP-seq bar the IP itself. It is also often referred to as “input”, a less precise term as it may refer to other kinds of ChIP-seq controls. We apologise – we accidentally omitted this acronym in the original submission. The Smchd1-GFP cells without a mutation in Smchd1 also serve as controls for the Smchd1-GFP-MommeD43 ChIP. We have incorporated these explanations more clearly into the text and thank the reviewer for raising these points.

4) There is considerable information in the literature regarding a crosstalk between SMCHD1 and the CTCF insulator protein. Have the authors determined if CTCF binding is perturbed in the MommeD43 mutant?

Thank you for this suggestion. We have now performed CTCF ChIP-seq in NSCs that are null for Smchd1 or MommeD43, all on the C57BL/6 background. Here we have found relatively limited effects on CTCF binding in both null and MommeD43 samples. However, for the peaks that are significantly differentially enriched (determined by CSAW, Supplementary Table 10), binding is enhanced in the null samples and depleted in the MommeD43 samples (Fig. 6c). These data are consistent with the effects that we observe on gene expression on autosomes and insulation on the inactive X. They are inconsistent with the effects on chromatin architecture observed by *in situ* HiC and DNA FISH. These data are therefore consistent with our conclusion that the MommeD43 mutation impacts functions of SMCHD1 that do not relate to chromatin architecture.

REVIEWERS' COMMENTS

Reviewer #2 (Remarks to the Author):

The authors have carefully addressed each of the comments and concerns in my original review. They have not demonstrated any of these mechanisms in FSHD patient and control cells, however have tone down the translational of these results to be something of potential. The manuscript is largely focus on gene function and mechanism so it's fair that this work can be done in the future.

Reviewer #3 (Remarks to the Author):

The authors have addressed the questions that were raised during review. Additional data were generated to answer some critical points. I have no further concerns.